# Altered tRNA dynamics during translocation on slippery mRNA as determinant of spontaneous ribosome frameshifting

Panagiotis Poulis[1], Anoshi Patel[1,2], Marina V. Rodnina[1] & Sarah Adio ⊕ [2✉]

When reading consecutive mRNA codons, ribosomes move by exactly one triplet at a time to synthesize a correct protein. Some mRNA tracks, called slippery sequences, are prone to ribosomal frameshifting, because the same tRNA can read both 0- and –1-frame codon. Using smFRET we show that during EF-G-catalyzed translocation on slippery sequences a fraction of ribosomes spontaneously switches from rapid, accurate translation to a slow, frameshifting-prone translocation mode where the movements of peptidyl- and deacylated tRNA become uncoupled. While deacylated tRNA translocates rapidly, pept-tRNA continues to fluctuate between chimeric and posttranslation states, which slows down the re-locking of the small ribosomal subunit head domain. After rapid release of deacylated tRNA, pept-tRNA gains unconstrained access to the –1-frame triplet, resulting in slippage followed by recruitment of the –1-frame aa-tRNA into the A site. Our data show how altered choreography of tRNA and ribosome movements reduces the translation fidelity of ribosomes translocating in a slow mode.

[1] Department of Physical Biochemistry, Max Planck Institute for Multidisciplinary Sciences, Göttingen, Germany. [2] Department of Molecular Structural Biology, Institute for Microbiology and Genetics, Georg-August University of Göttingen, Göttingen, Germany. ✉email: sarah.adio@uni-goettingen.de

During translation elongation, the ribosome moves along the mRNA in steps of three nucleotides, i.e., one codon at a time, towards the mRNA 3' end. The exact step size is essential, as it maintains the correct open reading frame until the ribosome encounters a stop codon that terminates translation. However, in some cases, ribosomes can switch from the original open reading frame (0 frame) into alternative −1, +1, +2 or −4 frames. Frameshifting usually occurs on "slippery" mRNA sequences which allow tRNAs to base pair with codons in both the original and an alternative frame. In programmed ribosome frameshifting, which is most prevalent in viruses, additional secondary structure elements in the mRNA facilitate frameshifting, resulting in two functional protein products that are usually essential for virus propagation[1–7]. In contrast, in most cases where slippage is spontaneous, −1 frameshifting results in out-of-frame decoding and termination at premature stop codons leading to synthesis of aberrant dysfunctional proteins[8–10]. The genomic abundance of slippery sequences is relatively high. In *Escherichia coli*, the slippery A AAA AAG sequence is found in 68 genes[11] and in humans ~10% of all cellular mRNAs contain a slippery sequence[12]. However, the frequency of spontaneous frameshifting in vivo is as low as $10^{-4}$–$10^{-5}$ during canonical translation[13,14], even though some slippery sequences actually favor codon-anticodon pairing in the −1 frame[15]. This implies existence of a mechanism preventing the loss of the correct reading frame at slippery sites. Recent studies demonstrated that elongation factor G (EF-G) has a key role in reading frame maintenance during tRNA–mRNA translocation[16–19].

Translocation is a complex process that entails movements of the ribosome, tRNAs, and mRNA. Pretranslocation (PRE) complexes comprise ribosomes with the mRNA-bound peptidyl-tRNA in the A site and deacylated tRNA in the P site. PRE complexes are dynamic and interconvert between two conformational states termed the non-rotated/classical state (or ground state 1) and the rotated/hybrid state (or ground state 2)[20–22]. In the classical state, tRNAs reside in the A/A, P/P sites, respectively, both on the small (SSU) and large (LSU) ribosomal subunits. In the rotated/hybrid state, the SSU rotates with respect to the LSU and tRNAs move into hybrid (A/P, A/P*, and P/E) conformations, in which the tRNA anticodons remain in the A and P site, respectively, while the acceptor arms move towards the P and E site on the LSU[19–21,23–37]. The elbow region of the A/P-site tRNA is mobile and can adopt slightly different orientations, A/P and A/P*[19,24,25]. EF-G binding facilitates the movement of tRNAs into the hybrid states, as well as rotation of the ribosomal subunits relative to each other[21,23,30,38–42]. GTP hydrolysis by EF-G and the subsequent Pi release promote large-scale rearrangement of the complex that uncouples the movements of the head and body domains of the SSU. The tRNA anticodons and the mRNA move with the SSU head domain in forward direction, whereas the SSU body domain rotates backwards relative to LSU. As a result, tRNAs move from the hybrid into intermediate states called chimeric states (CHI or ap/P, pe/E), where the anticodons bind between the A and P and P and E sites on the SSU and the acceptor ends reside in the P and E site of the LSU, respectively[19,23–25,38,41,43–45]. Extensive analyses by time-resolved cryo-EM, single-molecule, and ensemble kinetics have shown that CHI states are authentic intermediates of translocation[19,23,25,46]. Subsequently, tRNAs move synchronously to post translocation states (P/P and E/E)[38,41]. At the end of translocation, the deacylated tRNA and EF-G dissociate from the ribosome and the SSU head domain swivels backwards locking pept-tRNA in the P site[25,38,41,43].

Structural studies and kinetic analysis of translocation suggest how EF-G contributes to the reading frame maintenance. When ribosomes enter the CHI state, the stabilizing interactions between the tRNA–mRNA complex and 16 S rRNA on the SSU body domain are lost and instead the complex interacts with the residues at the tip of EF-G domain 4, in particular H583 and Q507[19,25,45]. In the absence of EF-G, the movements of the mRNA and tRNA anticodon become uncoupled and may lead to frameshifting[18]. Amino acid replacements at the tip of EF-G domain 4 promote −1 frameshifting on a slippery mRNA[16,17]. These replacements slow down translocation[16,23,47] and alter the timing of translocation events, i.e., the backward movement of the SSU head from the swiveled to the non-swiveled state is delayed and uncoupled from the release of the deacylated tRNA from the E site[16]. These two steps occur simultaneously during translocation with the wild-type (wt) EF-G[38,41] suggesting that rapid translocation and the EF-G-coordinated order of movements prevent spontaneous frameshifting.

While previous studies offered a conceptual framework for understanding reading frame maintenance, it remains unclear when and how spontaneous frameshifting occurs. Here we use smFRET to compare tRNA translocation pathways in the absence of frameshifting, i.e., on non-slippery mRNA, and at conditions where a significant fraction of ribosomes changes from 0 to −1 frame, i.e., on a slippery mRNA and with frameshifting-promoting EF-G mutants. This approach allows us to identify potential heterogeneity within the ribosome population, to visualize local tRNA fluctuations, and to monitor how internal movements of the ribosome control frameshifting. We identify two different modes of ribosome progression along the mRNA, a fast and accurate mode, where rapid simultaneous movement of pept- and deacylated tRNA is coupled to the motions of the ribosome, and a slow, frameshifting-prone mode with uncoupled translocation of pept- and deacylated tRNA. In the slow mode, pept-tRNA is trapped in fluctuations between CHI and P/P states before accommodating in the P site, which occurs during back swivel of the SSU head domain. Deacylated tRNA translocates at normal speed and dissociates from the ribosome before the accommodation of pept-tRNA in the P site is completed. This opens a time window where a single pept-tRNA can reassign the reading frame according to the thermodynamic potential of the mRNA sequence. Ribosomes favor the fast mode during translocation with wild-type EF-G, but switch to a slow mode during translocation by frameshifting-promoting EF-G mutants, which suggests that a population of ribosomes with distinct kinetic features contributes to the loss of reading frame on a slippery mRNA.

## Results

**Translocation of peptidyl-tRNA and ribosome frameshifting.** To dissect the mechanism of ribosome slippage, we first used smFRET to monitor EF-G-induced translocation of pept-tRNA from the A to the P site (Fig. 1a). We assembled PRE complexes on mRNA with or without a slippery sequence (Fig. 1b) and utilized FRET reporters attached to the ribosomal protein L11 (L11-Cy3) and pept-tRNA$^{Lys}$-Cy5 to follow pept-tRNA movements. On the non-slippery mRNA, tRNAs fluctuate between PRE states with FRET 0.8 and 0.6 until EF-G binds and then rapidly move to the P/P POST state (FRET 0.2) (Fig. 1c, d). Steady-state smFRET experiments (Supplementary Fig. 1) and distance measurements based on the recent cryo-EM structures[19,24,25] show that pept-tRNA in A/A and A/P states contribute to the FRET 0.8 state, whereas FRET 0.6 is consistent with the A/P* state that forms as the pept-tRNA elbow region and the CCA end move towards the P site. The transition from PRE to POST state occurs within 33 ms, i.e., one camera frame in our experimental setup. The estimated lower limit for the translocation rate ($k_{TL} \geq 30\,\text{s}^{-1}$) agrees with ensemble in vitro

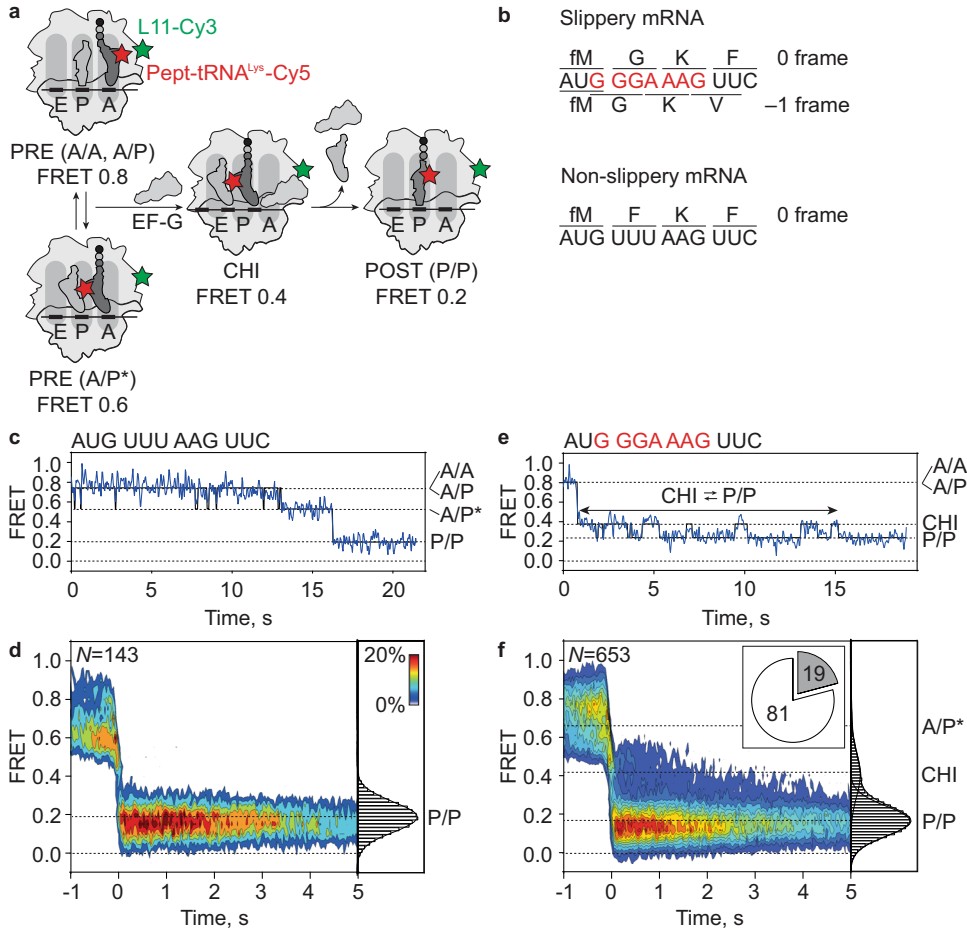

**Fig. 1 EF-G-induced translocation of pept-tRNA. a** Schematic of the smFRET experiment. Pept-tRNA[Lys]-Cy5 (red star) fluctuates between A/A, A/P, and A/P* states in the absence of EF-G. Upon EF-G addition, pept-tRNA moves into a transient CHI state before reaching the P/P state. The movement is monitored by change in FRET between pept-tRNA[Lys]-Cy5 and L11-Cy3 (green star). FRET values are assigned to PRE and POST states in independent experiments (Supplementary Fig. 1) and validated by distance measurements based on cryo-EM structures[19,24,25]. **b** Coding sequences of mRNA constructs. Slippery sequence encodes fMet-Gly-Lys-Phe peptide in 0 frame and fMet-Gly-Lys-Val in −1 frame. Non-slippery mRNA that does not support frameshifting encodes only the 0-frame peptide fMet-Phe-Lys-Phe. **c** Representative smFRET time trace showing tRNA translocation on non-slippery mRNA. Trace shows fluctuations between FRET 0.8 (A/A, A/P) and 0.6 (A/P*) followed by rapid transition to FRET 0.2 (P/P). Black line shows the HMM fit of the data here and in all smFRET traces. **d** Contour plot showing distribution of FRET values during tRNA translocation on non-slippery mRNA. Transitions occur from either A/A and A/P or A/P* to P/P in less than 33 ms. Traces are synchronized to the first transition below FRET 0.5. Histogram at the right shows distribution of FRET values after synchronization. Data are from 5 independent experiments ($N = 5$). **e** Representative smFRET time trace showing tRNA translocation on slippery mRNA. Trace shows rapid transition from FRET 0.8 (A/A, A/P) to FRET 0.4 (CHI)[23,46], followed by fluctuations between FRET 0.4 and 0.2 before adopting a long-lived FRET 0.2 state (P/P). **f** Contour plot showing distribution of FRET values during translocation on slippery mRNA. The contour plot contains mixture of trajectories. 81% of traces show direct transitions from A/A, A/P or A/P* to P/P. 19% of traces show transitions from A/A, A/P, or A/P* to CHI followed by fluctuations between CHI and P/P states before transition to long-lived P/P state. Traces are synchronized to the first transition below FRET 0.5. Histogram at the right shows distribution of FRET values after synchronization. Data are from 12 independent experiments ($N = 12$).

translocation rates ranging between 7 and 30 s[−1] at 22 °C[48–50]. On the slippery mRNA, most ribosomes (81%) show tRNA trajectories similar to those on non-slippery mRNA. However, on a fraction of ribosomes (19%), the translocation trajectory appears different and pept-tRNA[Lys]-Cy5 transiently samples FRET 0.4 before reaching P/P (Fig. 1e, f). Previously, we and others have shown that FRET 0.4 corresponds to the CHI (ap/P) state where the anticodon stem-loop of the tRNA resides between A and P site on the SSU[23,44–46]. During translocation on non-slippery mRNA, the CHI state is too transient to be captured (Fig. 1c, d). On the slippery mRNA, the movement of pept-tRNA[Lys]-Cy5 into CHI is rapid, but the exit from the CHI state is delayed due to enduring fluctuations between the CHI and P/P states before completing translocation (Fig. 1e, f).

We then performed analogous experiments using the slippery mRNA and EF-G mutants carrying single amino acid substitutions of residue Q507 at the tip of domain 4, EF-G(Q507A), EF-G(Q507N), and EF-G(Q507D) (Fig. 2a). Replacements of Q507 promote −1 frameshifting and the frameshifting efficiency depends on the type of amino acid substitution[16]. With different mutants, we again find two ribosome populations, one where tRNAs move rapidly from PRE to POST and another in which tRNAs dwell in transitions between CHI and P/P states before reaching the POST state. The percentage of ribosomes with a delayed exit from the CHI state increases in the order EF-G(Q507A) < EF-G(Q507N) < EF-G(Q507D) (Fig. 2b–d). The variation in the fraction of rapidly translocating ribosome population is not due to ribosome binding defects of EF-G

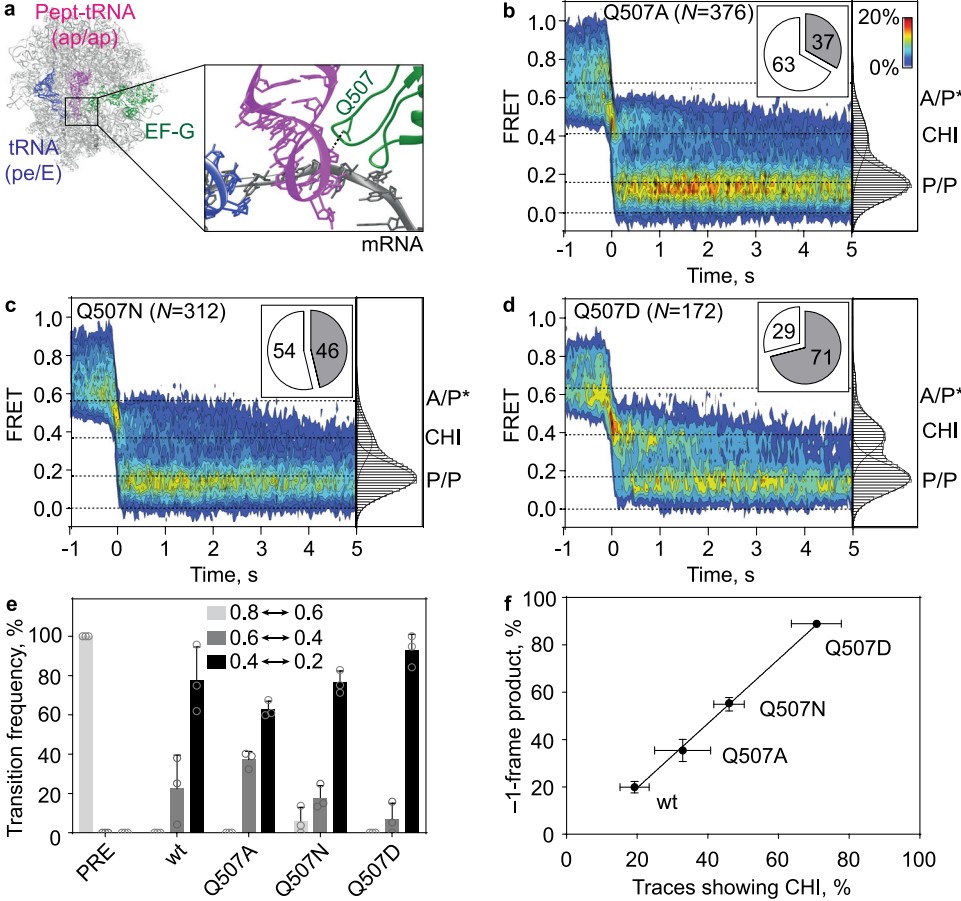

**Fig. 2 Frameshifting-prone translocation with EF-G(Q507) mutants. a** Schematic of an EF-G–ribosome complex in CHI state (adapted from PDB 4W29[45]). Pept-tRNA (magenta) and deacylated tRNA (blue) are in CHI states. The zoom-in shows residue Q507 at the tip of domain 4 of EF-G (green) in contact with the anticodon loop of pept-tRNA. The ribosome is shown in gray and the mRNA in black. **b–d** Contour plots showing the distribution of FRET values during translocation on slippery mRNA mediated by EF-G(Q507A) (b), EF-G(Q507N) (c), and EF-G(Q507D) (d). Histograms at the right show distribution of FRET values after synchronization to the first transition below FRET 0.5. Pie charts indicate percentage of smFRET traces showing prolonged fluctuations between CHI and P/P states (dark gray) during translocation. Data are from at least four independent experiments ($N = 4$ for Q507A, $N = 5$ for Q507N, and $N = 9$ for Q507D). **e** Transition frequency between FRET states during translocation on slippery mRNA with Q507 mutants and EF-G(wt) (Supplementary Fig. 2 and Table 1). After addition of EF-G, transitions between FRET 0.4 (CHI) and 0.2 (P/P) states are predominant. Shown are mean values with error bars representing the standard deviations. Data are from at least three independent experiments ($N = 3$ for PRE, $N = 12$ for wt, $N = 4$ for Q507A, $N = 5$ for Q507N, and $N = 9$ for Q507D). **f** Correlation between frameshifting efficiency and the fraction of ribosomes sampling CHI states during translocation on slippery mRNA (Supplementary Fig. 3). Frameshifting efficiencies were measured at 22 °C to match the conditions of the smFRET experiments. Shown are mean values with error bars representing the standard deviation. Black line indicates a linear fit with the slope of $1.3 \pm 0.1$, $R^2 = 0.9982$. Frameshifting efficiencies are from three independent experiments ($N = 3$). The percentage of traces with CHI states is derived from at least three independent experiments ($N = 12$ for wt, $N = 4$ for Q507A, $N = 5$ for Q507N, and $N = 9$ for Q507D).

mutants[16]. EF-G wt and all Q507 variants stabilize the A/P* state upon binding to PRE prior to translocation (Supplementary Fig. 2a) and the transitions PRE to the CHI state are rapid in all cases ($k_{TL} \geq 30\,\text{s}^{-1}$, Fig. 1e). The prevalent fluctuations on the slowly translocating ribosomes are between CHI (FRET 0.4) and P/P (FRET 0.2) states (Fig. 2e, Supplementary Fig. 2b, Table 1). The overall translocation rate on ribosomes delayed in the CHI state is low ($\sim 0.2\,\text{s}^{-1}$) and similar with EF-G(wt) and Q507 mutants (Supplementary Fig. 2c, Table 1), which is two orders of magnitude slower compared to ribosomes that transit the CHI state rapidly. Notably, the biochemical frameshifting efficiency (determined in independent experiments by quantitative analysis of peptide products; Supplementary Fig. 3) correlates very well with the slow population among all translocating ribosomes (Fig. 2b–d, f). The slope of the linear regression is close to 1, suggesting that frameshifting is to a large extent due to the fraction of slow ribosomes. These findings indicate that

ribosomes slip into the −1 frame while pept-tRNA is in the CHI state or upon attempting to move from CHI to the POST state.

To verify that slowly translocating ribosomes are prone to spontaneous frameshifting, we monitored in the same experiment translocation and aa-tRNA binding to the next A-site codon. We followed translocation of pept-tRNA$^{Lys}$ labeled by a quencher BHQ2 (pept-tRNA$^{Lys}$-BHQ2) relative to L11-Cy3 (Methods) and binding of the 0-frame Phe-tRNA$^{Phe}$-Cy5 or the −1-frame Val-tRNA$^{Val}$-Cy5, respectively, by FRET with L11-Cy3 (Fig. 3). Binding of Lys-tRNA$^{Lys}$-BHQ2 to L11-Cy3-labeled POST complexes leads to the drop of Cy3 fluorescence due to quenching (Supplementary Fig. 4a, b). The resulting PRE1 complex shows fluctuations between two states with low fluorescence intensity, presumably A/A, A/P and A/P* conformations (Supplementary Fig. 4b). On the non-slippery mRNA, PRE1 rapidly (within one frame) converts into POST2 upon addition of EF-G, leading to the

**Table 1 Kinetics of pept-tRNA$^{Lys}$ fluctuations during translocation on slippery mRNA.**

| EF-G | $k_{TL}$, s$^{-1}$ (N) | FRET, μ±s.d. | Transition rates during translocation $k$, s$^{-1}$ ($n$) | |
|---|---|---|---|---|
| | | | 0.6 → 0.4<br>0.4 → 0.6 | 0.4 → 0.2<br>0.2 → 0.4 |
| wt | 0.2 ± 0.1 (102) | 0.6 ± 0.1 | n.d.[a] (64) | 2.6 ± 0.6 (186) |
| | | 0.4 ± 0.1 | n.d. (61) | 2.7 ± 1.4 (148) |
| | | 0.2 ± 0.1 | | |
| Q507A | 0.1 ± 0.1 (110) | 0.6 ± 0.1 | 3.9 ± 0.7 (79) | 2.9 ± 1.2 (152) |
| | | 0.4 ± 0.1 | 5.8 ± 1.9 (83) | 2.8 ± 0.6 (138) |
| | | 0.2 ± 0.1 | | |
| Q507N | 0.2 ± 0.1 (106) | 0.6 ± 0.1 | n.d. (61) | 3.6 ± 0.6 (261) |
| | | 0.4 ± 0.1 | n.d. (64) | 3.7 ± 1.1 (228) |
| | | 0.2 ± 0.1 | | |
| Q507D | 0.3 ± 0.1 (90) | 0.6 ± 0.1 | n.d. (22) | 2.7 ± 0.2 (324) |
| | | 0.4 ± 0.1 | n.d. (21) | 3.6 ± 0.2 (315) |
| | | 0.2 ± 0.1 | | |
| wt–GTPγS | 0.2 ± 0.1 (128) | 0.6 ± 0.1 | 7.0 ± 0.9 (292) | n.d. (9) |
| | | 0.4 ± 0.1 | 6.2 ± 0.9 (266) | n.d. (9) |
| | | 0.2 ± 0.1 | | |
| wt–Spc | n.a.[b] | 0.8 ± 0.1 | 3.5 ± 0.4 (809)[c] | n.d. (39) |
| | | 0.6 ± 0.1 | 6.1 ± 1.2 (939)[d] | n.d. (29) |
| | | 0.4 ± 0.1 | 5.2 ± 1.2 (937) | |
| | | 0.2 ± 0.1 | 2.4 ± 0.2 (870) | |

*N* number of traces.
μ mean of the Gaussian distribution.
*n* number of transitions.
All rates were corrected for photobleaching of the FRET dyes (Methods) and are shown as mean±s.d. with R$^2$ ≥ 0.96 in all cases.
[a]number of fluctuations was too small to calculate transition rates.
[b]The majority of traces did not reach the POST state and the $k_{TL}$ rate was limited by the photobleaching of the FRET dyes.
[c]Transitions from FRET 0.8 to 0.6.
[d]Transitions from FRET 0.6 to 0.8.

dequenching of the Cy3 fluorophore (Supplementary Fig. 4c, d) and presenting the 0-frame Phe codon in the A site. When Phe-tRNA$^{Phe}$-Cy5 binds to POST2 complexes (Supplementary Fig. 5a–d), we observe a two-state FRET efficiency change corresponding to ribosome binding (RB), a state previously assigned as codon reading, followed by the accommodation of Phe-tRNA$^{Phe}$-Cy5 in the A site[51] forming the PRE2 complex (Supplementary Fig. 5c–d). The accommodated Phe-tRNA$^{Phe}$ rapidly reacts with the pept-tRNA in the P site to form a peptide bond, resulting in a deacylated tRNA$^{Lys}$ in the P site and a pept-tRNA$^{Phe}$ in the A site. FRET population distribution analysis suggests that pept-tRNA$^{Phe}$ samples A/A, A/P and A/P* states (Supplementary Fig. 5d) in a similar manner as seen for pept-tRNA$^{Lys}$ (Supplementary Fig. 1a–c). PRE2 converts to POST3 complex upon translocation by EF-G(wt) (Supplementary Fig. 4e; in the example shown in Supplementary Fig. 5b, c Cy5 photobleaches result in a stable high Cy3 signal). In contrast, when Val-tRNA$^{Val}$-Cy5 complementary to the –1-frame codon is added to POST2 complexes formed on non-slippery mRNA, we observe only short-lived FRET event (Supplementary Fig. 5e–h), showing Val-tRNA$^{Val}$ as it makes multiple unsuccessful attempts to read a near-cognate 0-frame codon[51]. The dwell time of Val-tRNA$^{Val}$ on POST2 complexes is two orders of magnitude lower than the dwell time of cognate Phe-tRNA$^{Phe}$ ($k_{off}$(Val-tRNA$^{Val}$) = 22 s$^{-1}$, $k_{off}$(Phe-tRNA$^{Phe}$) = 0.2 s$^{-1}$, Supplementary Fig. 6a). The lack of Val-tRNA$^{Val}$ accommodation suggests that on the non-slippery mRNA and with EF-G(wt), the ribosomes remain in the 0-frame.

When we carry out the same experiment on slippery mRNA with EF-G(wt), on the majority of ribosomes, pept-tRNA$^{Lys}$-BHQ2 moves rapidly from PRE1 into POST2 (Fig. 3a–d). –1-frame Val-tRNA$^{Val}$ dissociates rapidly ($k_{off}$(Val-tRNA$^{Val}$) = 18 s$^{-1}$, Supplementary Fig. 6b). Rapid rejection of Val-tRNA$^{Val}$ indicates that the reading frame is maintained after rapid translocation. However, on a minor fraction of ribosomes pept-tRNA$^{Lys}$-BHQ2 rapidly moves from PRE1 into a state with intermediate fluorescence intensity between PRE1 and POST2 (Fig. 3e, f), which has the same decay rate as the CHI state observed with the L11-Cy3/pept-tRNA$^{Lys}$-Cy5 FRET pair ($k_{off}$(CHI) = 0.2 s$^{-1}$, Supplementary Fig. 7 and 2c). After slow translocation, –1-frame Val-tRNA$^{Val}$-Cy5 is accommodated into POST2 and then fluctuates between A/A, A/P and A/P* states (Fig. 3f–h). On these complexes, the dwell time of tRNA$^{Val}$-Cy5 on PRE2 complexes is long ($k_{off}$(pept-tRNA$^{Val}$) = 0.1 s$^{-1}$, Supplementary Fig. 6c) and in the same range as the dwell time of 0-frame Phe-tRNA$^{Phe}$ on the non-slippery mRNA (Supplementary Fig. 6a). With EF-G(Q507D), we observe the accommodation of –1-frame Val-tRNA$^{Val}$-Cy5 on PRE2 complexes after slow translocation of pept-tRNA$^{Lys}$-BHQ2 on the majority of ribosomes (Fig. 3i–l, Supplementary Fig. 6c and 7). We did not observe the stable binding of –1-frame Val-tRNA$^{Val}$-Cy5 on PRE2 complexes after rapid translocation of pept-tRNA$^{Lys}$. These data show that after delayed translocation on the slippery mRNA by both EF-G(wt) and EF-G(Q507D) mutant, the ribosome moves into the −1 frame exposing the Val codon in the A site, whereas rapid translocation leads to 0 frame maintenance.

**Translocation of deacylated tRNA on slippery mRNA.** Next, we used smFRET between tRNA$^{Lys}$-Cy5 in the P site and the ribosomal protein S13-labeled with Cy3 (S13-Cy3) to follow the trajectory of deacylated tRNA movement from the P to the E site (Fig. 4a, b). In the PRE state on non-slippery mRNA, tRNA$^{Lys}$-Cy5 is found either in the P/P (FRET 0.9) or P/E (FRET 0.6) states (Supplementary Fig. 8). The tRNA fluctuates between the P/P and P/E states, albeit much less frequently (transition frequency 0.4 ± 0.1, Supplementary Fig. 8f) than pept-tRNA$^{Lys}$ between A/A, A/P, and A/P* states (transition frequency 5.5 ± 0.7, Supplementary Fig. 1c). Upon addition of EF-G, tRNA$^{Lys}$-Cy5 moves rapidly ($k_{TL} ≥ 30$ s$^{-1}$) from P/P or P/E into a

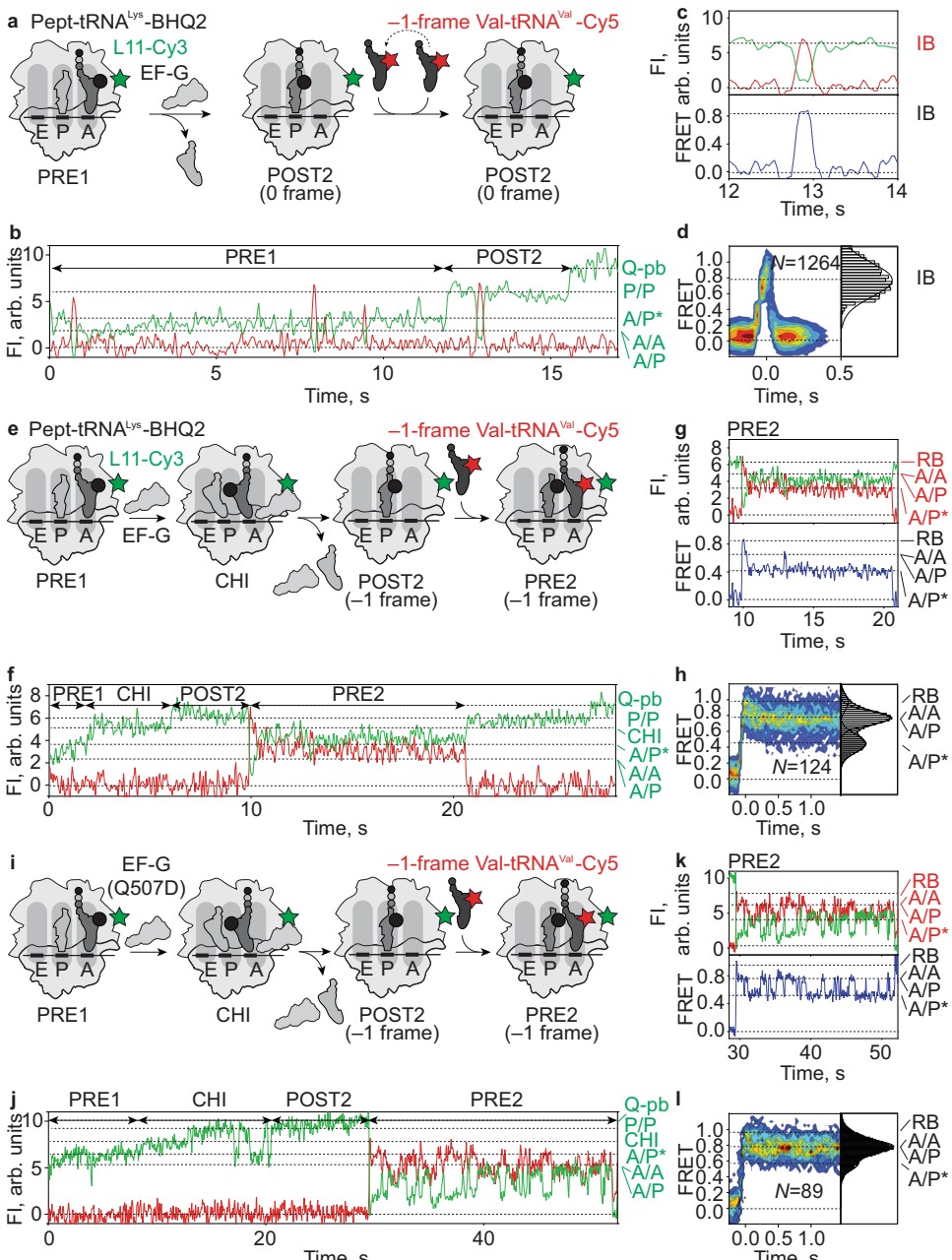

**Fig. 3 Translocation of pept-tRNA$^{Lys}$ on slippery mRNA and incorporation of 0- and –1-frame aa-tRNAs. a** Schematic of translocation on slippery mRNA with EF-G(wt). Pept-tRNA$^{Lys}$-BHQ2 (black circle) moves from the A to the P site. The near-cognate –1-frame Val-tRNA$^{Val}$-Cy5 (red star) samples POST2 complexes without accommodating in the A site. EF-G is added to immobilized PRE1 complexes together with the EF-Tu–GTP–Val-tRNA$^{Val}$-Cy5 complex. **b** Representative time trace of pept-tRNA$^{Lys}$-BHQ2 movement and subsequent sampling of POST2 complexes by Val-tRNA$^{Val}$-Cy5 ($N = 89$). FI, fluorescence intensity; Q-pb, photobleaching of BHQ2. Green labels at the right Y-axis indicate tRNA$^{Lys}$ conformational states monitored by Cy3 fluorescence, as assigned in Supplementary Fig. 4. **c** Zoom-in into **b** showing Cy3 and Cy5 FI and calculated FRET of the initial binding (IB) without accommodation of –1-frame Val-tRNA$^{Val}$-Cy5 on POST2 complexes. **d** Contour plot showing distribution of FRET values during –1-frame Val-tRNA$^{Val}$-Cy5 sampling of POST2 complexes. Traces were synchronized to the point with FRET > 0. Histogram at the right shows FRET distribution after synchronization. **e** Schematic of translocation on slippery mRNA by EF-G(wt), with −1 frameshifting. After translocation of pept-tRNA$^{Lys}$-BHQ2 (black circle) from the A to the P site and Val-tRNA$^{Val}$-Cy5 (red star) can bind to its cognate –1-frame codon. **f** Representative time trace of pept-tRNA$^{Lys}$-BHQ2 translocation and accommodation of Val-tRNA$^{Val}$-Cy5 ($N = 57$). **g** Zoom-in into **f** showing Cy3 and Cy5 FI and calculated FRET of Val-tRNA$^{Val}$-Cy5 binding to POST2 forming PRE2 complexes. Red labels indicate conformational states of tRNA$^{Val}$-Cy5. **h** Contour plot showing distribution of FRET values after Val-tRNA$^{Val}$-Cy5 accommodation on POST2 complex forming PRE2. **i** Schematic of translocation on slippery mRNA by EF-G(Q507D), with −1 frameshifting. After translocation of pept-tRNA$^{Lys}$-BHQ2 from the A to the P site, Val-tRNA$^{Val}$-Cy5 (red star) accommodates on its cognate codon in –1-frame. **j** Representative time trace of pept-tRNA$^{Lys}$-BHQ2 translocation and subsequent accommodation of –1-frame Val-tRNA$^{Val}$-Cy5 (box) ($N = 46$). **k** Zoom-in into **k** showing Cy3 and Cy5 FI and calculated FRET of Val-tRNA$^{Val}$-Cy5 binding to POST2 complex, leading to CR and subsequent fluctuations between A/A, A/P and A/P* states. **l** Contour plot showing the distribution of FRET values after Val-tRNA$^{Val}$-Cy5 accommodation on POST2 complex.

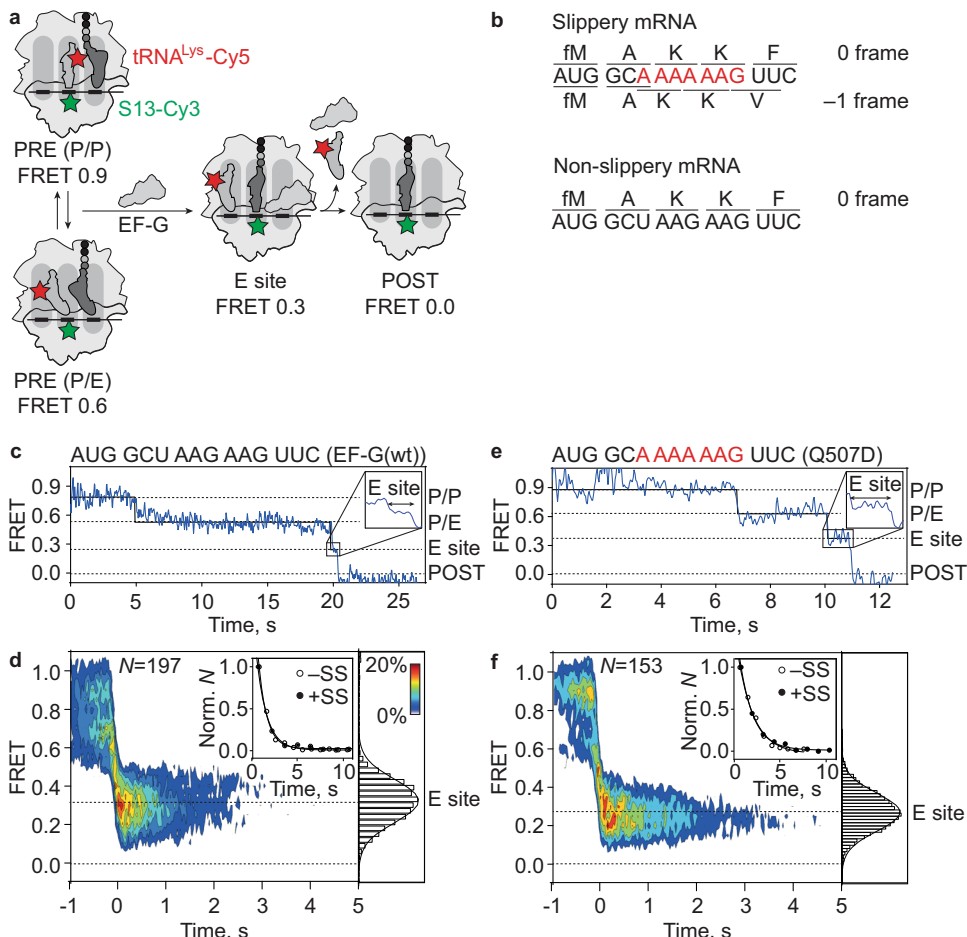

**Fig. 4 Translocation trajectory of deacylated tRNA. a** Schematic of smFRET experiment monitoring movement of tRNA$^{Lys}$-Cy5 (Cy5; red star) relative to protein S13-Cy3 (green star). Translocation is induced by addition of EF-G to immobilized PRE complexes. FRET values corresponding to P, P/E and E states were determined in independent experiments (Supplementary Fig. 5 and 6). **b** Coding sequences of mRNA constructs. Slippery sequence encodes fMet-Ala-Lys-Lys-Phe in 0 frame and fMet-Ala-Lys-Lys-Val in −1 frame. Non-slippery mRNA encodes only 0-frame fMet-Ala-Lys-Lys-Phe peptide because tRNA$^{Lys}$ does not base pair with the −1-frame GAA. **c** Representative smFRET time trace of tRNA translocation on non-slippery mRNA in the presence of EF-G(wt) showing step-wise transition from FRET 0.9 (P/P) to 0.6 (P/E) to 0.3 (E site) to 0.0 (dissociation). **d** Contour plot showing distribution of FRET efficiencies during translocation on non-slippery mRNA by EF-G(wt). Transitions occur either from FRET 0.9 (P site) or 0.6 (P/E) to FRET 0.3 (E site). Traces are synchronized to the first transition below FRET 0.5. Histogram at the right shows distribution of FRET values after synchronization. Inset shows rates and curve fits of tRNA dissociation from E site on slippery (closed circles) and non-slippery (open circles) mRNA (Supplementary Fig. 6b and Table 2). Normalization was performed by division by the number of transitions (n). Data are from four independent experiments (N = 4). **e** Representative smFRET time trace of tRNA translocation on slippery mRNA in the presence of EF-G(Q507D) showing step-wise transition from FRET 0.9 (P site) to 0.6 (P/E) to 0.3 (E site) to 0.0 (dissociation), similar to EF-G(wt) (**c**). **f** Contour plot showing the distribution of FRET efficiencies during translocation on slippery mRNA by EF-G(Q507D). Transitions occur either from FRET 0.9 (P site) or 0.6 (P/E) to FRET 0.3 (E site). Inset shows rates and curve fits of tRNA dissociation from E site on slippery (closed circles) and non-slippery (open circles) mRNA. Normalization was performed by division by the number of transitions (n) (Supplementary Fig. 6c and Table 2). Data are from three independent experiments (N = 3).

FRET 0.3 state followed by the loss of FRET (Fig. 4c, d). We assign FRET 0.3 to the transient occupancy of the E site by deacylated tRNA; the subsequent transition to FRET 0 reflects the dissociation from the ribosome (Fig. 4a, Supplementary Fig. 9a). The translocation trajectory shows no apparent CHI state for the P-site tRNA. The overall reaction of tRNA translocation and dissociation is rapid ($k_{off} = 1\,s^{-1}$, Fig. 3b–d, Table 2), in agreement with previous reports[38,41]. Surprisingly, we observed the same translocation pattern on slippery and non-slippery mRNA with rapid transition of tRNA$^{Lys}$-Cy5 to the E site followed by dissociation from the ribosome (Fig. 4d inset, Supplementary Fig. 9b, d, Table 2). Moreover, translocation with EF-G(wt) is not different from that with EF-G(Q507D), the mutant promoting the highest frameshifting efficiency and resulting in the lowest ensemble pept-tRNA translocation rate[16,17] (Fig. 4e, f,

Supplementary Fig. 9c, d, Table 2). In all cases, translocation of deacylated tRNA is fast, directional, and irreversible. Notably, on ribosomes where pept-tRNA translocates slowly, translocation and subsequent dissociation of deacylated tRNA is much faster than the movement of pept-tRNA (Tables 1 and 2), i.e., translocation of pept-tRNA from the A to the P site is not yet completed when deacylated tRNA has already cleared the E site. This indicates that movements of pept-tRNA are uncoupled from the displacement of deacylated tRNA on the fraction of ribosomes that undergo frameshifting, allowing pept-tRNA to more easily access the −1-frame codon after the E-site tRNA release.

**Identification of key transitions leading to frameshifting.** To further narrow down the timing of spontaneous −1

**Table 2 Kinetics of deacylated tRNA$^{Lys}$ dissociation from the E site.**

| EF-G | Dissociation rate from E site, s$^{-1}$ ($N$)[a,b] | |
|---|---|---|
| | U AAG AAG (non-slippery) | A AAA AAG (slippery) |
| wt | 0.9 ± 0.1 (196) | 1.0 ± 0.1 (169) |
| Q507D | 0.6 ± 0.1 (175) | 0.5 ± 0.1 (152) |
| wt-GTPγS | 0.2 ± 0.1 (205) | 0.3 ± 0.1 (173) |
| wt-Spc | n.d. | 0.3 ± 0.1 (157) |
| wt-FA | <0.1 ± 0.1[c] (169) | n.d. |

[a]$N$, number of traces.
[b]All rates were corrected for photobleaching of the FRET dyes (Methods) and are shown as mean±s.d. from at least three independent experiments. $R^2 \geq 0.96$ in all cases.
[c]Rate was limited by the photobleaching of the FRET dyes.

frameshifting, we studied translocation on slippery mRNA in the presence of EF-G(wt) and GTPγS, a slowly hydrolyzable GTP analog that binds to translational GTPases with similar affinity and orientation as GTP[52]. Ensemble translocation rates are reduced by ~30-fold by replacing GTP with GTPγS, because tRNAs are trapped at the early steps of translocation (Fig. 5a)[23,38,53,54]. Although translocation rates with EF-G(wt)–GTPγS and EF-G(Q507D)–GTP are similar[16,17], the frameshifting efficiency on slippery mRNA is much lower than with EF-G(Q507D)–GTP, and is similar to that with EF-G(wt)–GTP (Fig. 5a). This result indicates that slow translocation alone does not explain the high frameshifting efficiency, which prompted us to analyze the pept-tRNA translocation trajectories in the presence of GTPγS (Fig. 5b–h). GTPγS did not affect the ability of EF-G to stabilize pept-tRNA in the A/P* state (Supplementary Fig. 10a). The majority of traces (73%) shows slow translocation while sampling the FRET 0.4 state (Fig. 5b, c, Table 1), which may be an authentic CHI or a CHI-like off-pathway state. Pept-tRNA fluctuates predominantly between A/P* and CHI, rather than between CHI and P/P states as observed with EF-G–GTP on slippery mRNA on the fraction of slow ribosomes (Fig. 5d, Supplementary Fig. 10b). The overall pept-tRNA translocation rate on a slippery mRNA is ~0.2 s$^{-1}$ (Table 1), similar to that measured with GTPγS on a non-slippery sequence, 0.17 s$^{-1}$[23]. Notably, also translocation of deacylated tRNA$^{Lys}$-Cy5 is slow (0.2–0.3 s$^{-1}$; Fig. 5e, f, Supplementary Fig. 10c, Tables 1 and 2), indicating synchronous displacement of the two tRNAs in the presence of EF-G–GTPγS. Deacylated tRNA dwells mostly in the E site, which may prevent the pept-tRNA from sampling the −1-frame codon and would explain why the reading frame is maintained.

We then used the antibiotic spectinomycin (Spc) as an alternative method to stall translocation at an early stage (Fig. 5a, Supplementary Fig. 11)[24,41]. Spc binds to h34 of 16 S rRNA connecting the head and the body domain of the SSU[24,55]. Our smFRET translocation experiments show that the effect of Spc is similar to that of EF-G–GTPγS: pept-tRNA fluctuates between A/P* and a CHI-like state, whereas the dwell time of deacylated tRNA in the E site is prolonged; both tRNAs do not proceed to POST states and the efficiency of −1 frameshifting is low (Fig. 5a, Supplementary Fig. 11). These results suggest that although Spc binding to the ribosome does not affect GTP hydrolysis by EF-G, Spc blocks the conformational rearrangements that couple GTP hydrolysis and Pi release to tRNA movement. With respect to the mechanism of −1 frameshifting, these findings indicate that spontaneous ribosome slippage occurs with pept-tRNA in CHI state or attempting to move from CHI to the POST state, when deacylated tRNA has been released from the ribosome.

**Movements of SSU head domain during translocation on slippery mRNA.** To test the potential effect of ribosome dynamics, we monitored the SSU head domain movements using a validated FRET pair with labels on the SSU protein S13 (S13-Cy3) and the LSU protein L33 (L33-Cy5) (Fig. 6a)[38,48]. On PRE complexes in the absence of EF-G, we observe two inter-converting states with FRET 0.5 and 0.8, respectively, corresponding to non-rotated (N) and rotated-swiveled (S) states of the SSU head domain (Supplementary Fig. 12a-d). POST complexes are predominantly in the FRET 0.5 (N) state (Supplementary Fig. 12e–g). After the addition of EF-G to PRE complexes, the majority of ribosomes transiently populate the S state before adopting a long-lived N (POST) state (85% on non-slippery and 87% on slippery mRNA; Fig. 6b, Supplementary Fig. 13a). Notably, back swiveling of the SSU head domain in the presence of EF-G occurs at a rate similar to that on PRE complexes in the absence of EF-G (Supplementary Figs. 12d, 13b and Table 3), suggesting that back swiveling may occur spontaneously after EF-G dissociation from the ribosome. The transition rates from the last S to long-lived N state are similar on non-slippery and slippery mRNA in the presence of EF-G(wt) (Supplementary Fig. 13b and Table 3) and agree with rates measured in previous ensemble and smFRET experiments[16,17,38,41,43]. The minor population of complexes (15% on non-slippery mRNA and 13% on slippery mRNA) stayed in the S state and did not reach the N state during the time of the experiment.

When we induced translocation on slippery mRNA by EF-G(Q507D), the ratio of ribosomes showing fast and slow back swiveling is inversed (13% fast and 87% slow, Fig. 6b, d). Notably, similar fractions of ribosomes show slow backward swiveling (Fig. 6d) and slow pept-tRNA translocation (Fig. 2d). The rate of SSU head domain closure (<0.1 s$^{-1}$, Supplementary Fig. 13c and Table 3) is lower than that of pept-tRNA translocation (0.3 s$^{-1}$, Table 1) or the dissociation of deacylated tRNA from the ribosome (0.6 s$^{-1}$, Table 2), suggesting that pept-tRNA continues to fluctuate between CHI and P/P states because the SSU remains in a swiveled conformation, while deacylated tRNA dissociates from the ribosome. It is important to note that, in contrast to the swiveling motion of the SSU head domain, the rotational movement of the SSU body domain with respect to the LSU most likely does not contribute appreciably to spontaneous frameshifting, as both forward and backward SSU body rotation precede the formation of the CHI state[19,25,38,56]. In summary, our data show that slow translocation by EF-G(Q507D) stalls the SSU head domain in the swiveled state which renders ribosomes susceptible to spontaneous frameshifting due to delayed re-locking of pept-tRNA in the P site.

## Discussion

**Choreography of translocation on a slippery mRNA.** The present data show how slippery sequences affect the choreography of translocation and how this leads to frameshifting. While on the non-slippery mRNA all ribosomes behave in a quasi-uniform way (within the time resolution of TIRF experiments) and translocate rapidly, on a slippery mRNA we identify two distinct ribosome populations, one that translocates rapidly, and another that is slow in completing pept-tRNA translocation (Fig. 7a, b). The fraction of such slow ribosomes in the population correlates with the frameshifting efficiency. On slow ribosomes, both tRNAs rapidly move from the A/P or A/P* and P/E into their respective CHI states. Deacylated tRNA is then rapidly released from the ribosome. Normally, pept-tRNA becomes locked in the P site upon backward movement of the SSU head domain. On the slippery sequence, the locking is delayed and pept-tRNA fluctuates between CHI and P/P states. If base pairing of the pept-

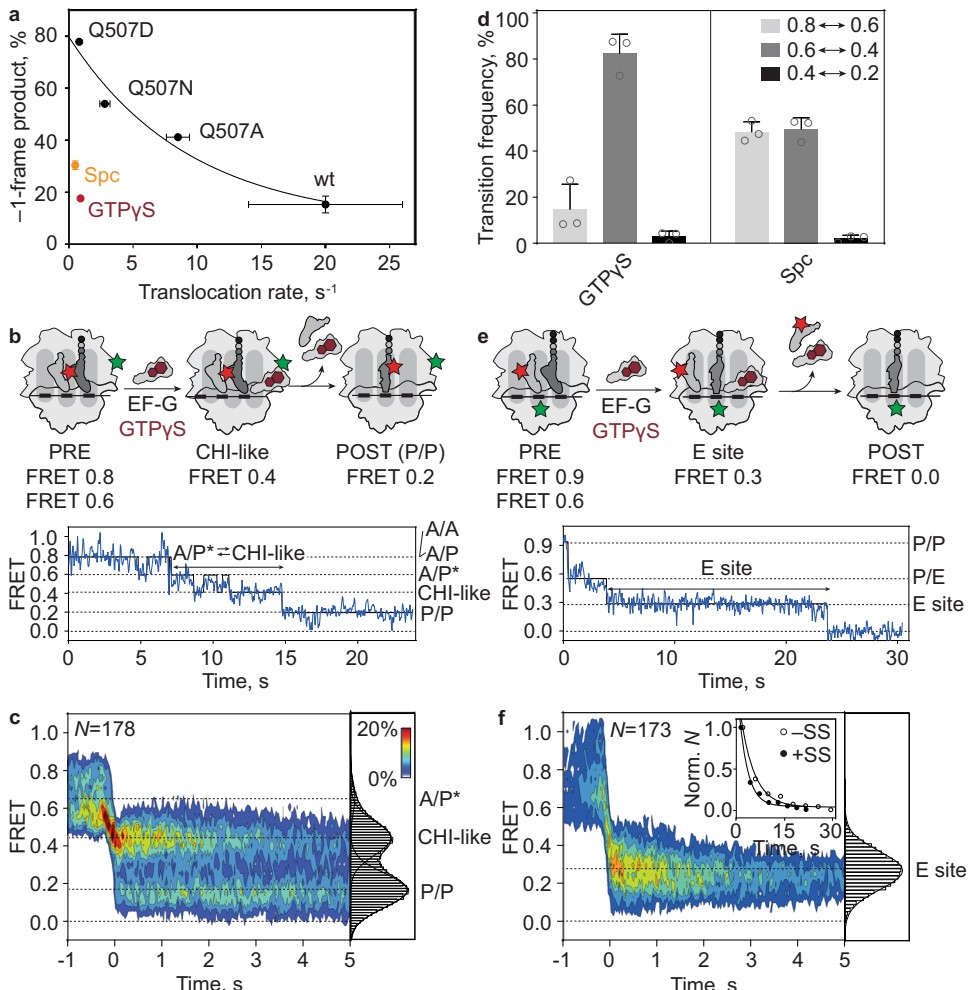

**Fig. 5 Translocation on slippery mRNA with EF-G(wt)—GTPγS. a** Correlation between frameshifting and ensemble translocation rate. Frameshifting was measured at 37 °C, data are presented as mean ± s.d. from three independent experiments ($N = 3$). Translocation rates of EF-G(wt) and Q507 mutants are from ref. [16], of GTPγS (brown) from ref. [54], and of Spc (orange) from [73]. **b** Above: schematic of smFRET experiment monitoring movement of pept-tRNA$^{Lys}$-Cy5 (red stars) relative to L11-Cy3 (green star) by EF-G(wt)—GTPγS (brown hexagon) binding to immobilized PRE complexes. Below: representative smFRET trace of pept-tRNA translocation by EF-G(wt)—GTPγS with fluctuations between FRET 0.8 (A/A, A/P) and 0.6 (A/P*) followed by fluctuations between FRET 0.4 (CHI) and 0.6. **c** Contour plot showing the distribution of FRET values during translocation of pept-tRNA on slippery mRNA by EF-G(wt)—GTPγS. Traces are synchronized to the first transition below FRET 0.5. Histogram at the right shows the distribution of FRET values after synchronization. Data are from six independent experiments ($N = 6$). **d** Transition frequencies between FRET states during pept-tRNA translocation on slippery mRNA by EF-G(wt)—GTPγS (Supplementary Fig. 7b and Table 1) and in the presence of Spc (Table 1). Data are presented as mean ± s.d. from 6 ($N = 6$, EF-G(wt)—GTPγS), or 3 ($N = 3$, Spc) independent experiments. **e** Above: schematic of smFRET experiment with tRNA$^{Lys}$-Cy5 (red star) moving relative to protein S13-Cy3 (green star) during translocation induced by addition of EF-G(wt)–GTPγS to immobilized PRE complexes. Below: representative smFRET trace of translocation by EF-G(wt)—GTPγS with step-wise transition from FRET 0.9 (P/P) to 0.6 (P/E) to 0.3 (E) to 0.0 (dissociation). **f** Contour plot showing the distribution of FRET values during translocation of deacylated tRNA on slippery mRNA by EF-G(wt)—GTPγS. Inset shows rates and curve fit of tRNA dissociation from E site on slippery (closed circles) and non-slippery (open circles) mRNA (Supplementary Fig. 7c and Table 2). Normalization was performed by division by the number of transitions ($n$). Data are from 3 independent experiments ($N = 3$).

tRNA anticodon with the −1-frame codon is favored over the 0-frame codon, as is often the case with slippery sequences[15], these continuing fluctuations provide the time window for pept-tRNA to switch to the −1-frame, followed by incorporation of the −1-frame tRNA.

Residues at the tip of EF-G domain 4, in particular Q507, are important for reading frame maintenance[16,17]. Recent cryo-EM structure suggests that in the A/P or A/P* state, EF-G domain 4 is flexible and most probably not involved in the stabilization of the codon-anticodon complex in the A site[19]. The codon-anticodon duplex is supported by the interactions with the residues of 16 S rRNA h44 in the SSU body domain[19,25]. Upon moving to the CHI state, the contacts with the SSU body domain are disrupted

and tRNA anticodon has a propensity to disconnect from its 0-frame codon[18,45]. EF-G residues at the tip of domain 4 stabilize the codon-anticodon duplex in its correct geometry, thereby contributing to the reading frame maintenance[18].

The timing of events during spontaneous frameshifting shows similarities but also differences to −1 programmed ribosome frameshifting (−1PRF)[57]. A hallmark of −1PRF is the slowing down of translocation at the slippery site facilitated by mRNA secondary structures, such as downstream stem loops or pseudoknots. During stalling, the SSU head domain is swiveled[58] and the SSU body is rotated with respect to the LSU[59,60], suggesting a key role of SSU dynamics both in programmed and spontaneous −1 frameshifting. Single-molecule studies of −1PRF

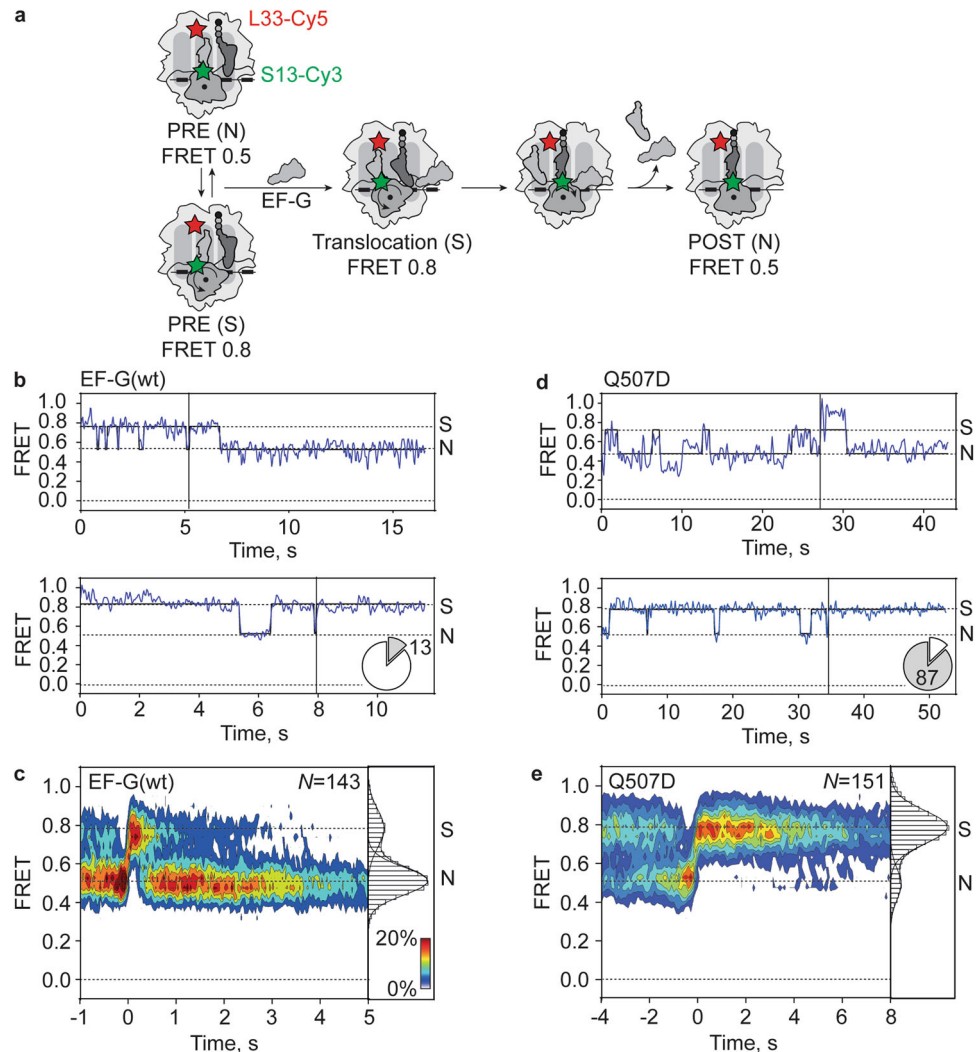

**Fig. 6 SSU head domain swiveling during translocation on slippery mRNA. a** Schematic of smFRET experiment. Movement of SSU head domain during translocation induced by addition of EF-G to immobilized PRE complexes monitored with FRET labels on ribosomal proteins S13 (S13-Cy3, green star) and L33 (L33-Cy5, red star). FRET values representing the non-swiveled (N) and swiveled (S) states are determined in independent experiments (Supplementary Fig. 9 and Table 3). **b** Representative smFRET time traces for SSU head movements during EF-G(wt)-induced translocation. The majority of traces (upper panel) show fluctuations between FRET 0.8 (S) and 0.5 (N) followed by a stable FRET 0.5 (N) state after translocation. 13% of traces (lower panel) show no transition to the N state in the time course of the experiment. Pie charts indicate the percentage of traces ending in N (white) or S (gray) state. Black vertical lines represent the synchronization point, i.e., the last transition to FRET 0.8 (S) (see below). **c** Contour plot showing distribution of FRET values representing SSU head movement during translocation on slippery mRNA by EF-G(wt). Traces were synchronized to the last transition to FRET 0.8 (S). The duration of the last FRET 0.8 state is an estimate for the duration of translocation, because back swiveling occurs simultaneously with the dissociation of EF-G from the ribosome after translocation[38]. Histogram at the right shows distribution of FRET values after synchronization. Data are from four independent experiments (*N* = 4). **d** Representative smFRET time traces for SSU head domain movement during translocation on slippery mRNA with EF-G(Q507D). A small fraction of traces (upper panel) show fluctuations between FRET 0.8 (S) and 0.5 (N) and end in a stable FRET 0.5 (N) state after translocation. 87% of traces (lower panel) show no transition to the N state in the time course of the experiment. **e** Contour plot showing distribution of FRET values representing SSU head domain movement during translocation on slippery mRNA by EF-G(Q507D). Histogram at the right shows the distribution of FRET values after synchronization. Data are from three independent experiments (*N* = 3).

on *E. coli dnaX* mRNA using optical tweezers showed that stalled ribosomes make multiple translocation attempts sampling sequences upstream or downstream of the 0 frame[61]. In our study, fluctuations of pept-tRNA between CHI and P/P states allow ribosomes to explore alternative reading frames and eventually re-equilibrate in a reading frame that is thermodynamically favored before resuming translation. The difference between the spontaneous slippage and –1PRF concerns the timing of deacylated tRNA release, which is rapid during spontaneous frameshifting (this paper and ref. [16]), but slow in the two examples of –1PRF where this was studied[58,62]. Thus, –1-

frameshifting can occur by different mechanisms with one (during spontaneous frameshifting) or two (during programmed frameshifting) tRNAs bound, provided pept-tRNA is trapped in fluctuations between CHI and P/P. –1-frameshifting can be also induced by the lack of the A-site tRNA, resulting in a "hungry" frameshifting[63], which is also a slow process, but proceeds via a different mechanism presumably involving spontaneous P-site tRNA slippage, rather than A-site tRNA translocation.

Tetrameric slippery mRNA sequences such as CC[C/U]-[C/U] can induce ribosome frameshifting in +1-direction when tRNAs either lack post-transcriptional modifications or contain

nucleotide insertions in their anticodon loops[64–66]. SufB2, a +1-frameshifting-prone tRNA mutant containing a G37a insertion in the anticodon loop of yeast *ProL* tRNA^Pro, uses triplet codon-anticodon-pairing in 0 frame, but shifts into the +1 frame in the process of translation. smFRET experiments using L1 stalk

movement as readout for translocation show that the transition of PRE to the POST state is much slower with SufB2 than with a canonical tRNA^Pro in the A site[66]. This suggests that slow tRNA movement correlates with +1 frameshifting, and hence slow translocation may be a hallmark for +1 or −1 frameshifting. It is not known which step of SufB2 translocation is slowed down, but cryo-EM suggests that the A-site tRNA shifts into the +1-frame soon after binding of EF-G–GDPCP[65]. If GDPCP stalls translocation at the same stage as GTPγS (Fig. 5 and refs. [23,38,41]), this would suggest that +1 frameshifting occurs early on the translocation pathway before CHI state formation, which would be clearly different to −1 frameshifting that occurs in the late phase of translocation after CHI state formation and during stabilization of the POST state.

**Slow gears of translocation.** Our results describe two alternative translocation pathways, one that ensures rapid coordinated movement of tRNAs in the correct reading frame, and the other which is slow and prone to −1 frameshifting (Fig. 7b). Recent force and fluorescence measurements suggested that ribosomes

| **Table 3 Kinetics of SSU head swiveling.** | | |
|---|---|---|
| **EF-G** | $k_{S \to N}$, s$^{-1}$ (n)[a,b] | |
| | **G UUU AAG (non-slippery)** | **G GGA AAG (slippery)** |
| no EF-G | n.d. | 2.5 ± 0.2 (1975) |
| wt | 2.2 ± 0.3 (149) | 1.9 ± 0.1 (124) |
| Q507D | n.d. | <0.1 ± 0.1[c] (151) |

[a]n, number of transitions.
[b]Rates are shown as mean±s.d. from at least three independent experiments. $R^2 \geq 0.99$ in all cases.
[c]Rate was limited by the photobleaching of the FRET dyes.

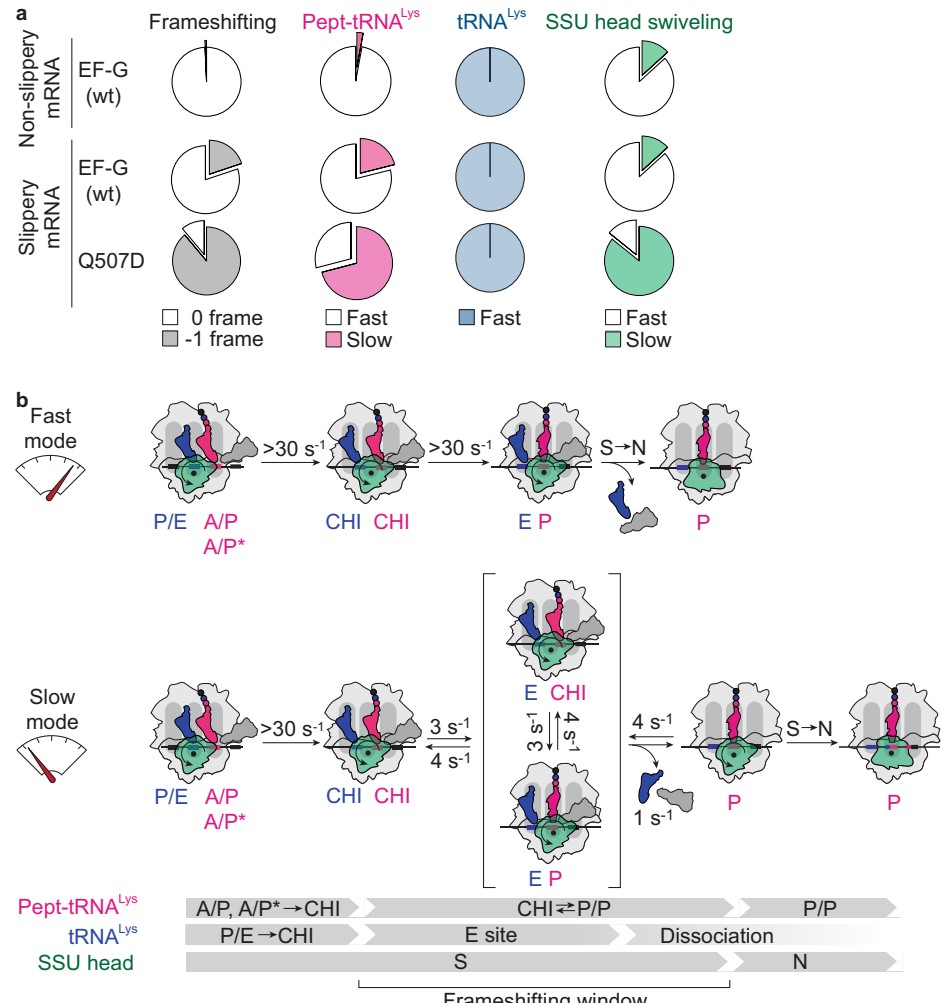

**Fig. 7 Translocation trajectories correlating with frameshifting. a** Pie charts comparing the frameshifting efficiency (gray) with the distribution of translocation rates for pept-tRNA (magenta) and deacylated tRNA (blue) and SSU head domain back rotation (green) on non-slippery and slippery mRNA with EF-G(wt) and EF-G(Q507D). **b** Kinetic model of translocation on slippery mRNA. Majority of ribosomes translocate in fast mode with tRNAs moving synchronously to the POST state; back swiveling of the SSU head domain completes translocation. A fraction of ribosomes translocates in slow mode where pept-tRNA is trapped fluctuating between CHI and P/P instead of moving to the POST state. Deacylated tRNA translocates rapidly and dissociates from the E site allowing pept-tRNA to sample 0- and –1-frame codons. SSU head closure is delayed due to prolonged fluctuations of pept-tRNA. Rates of the elemental reactions for pept-tRNA are indicated.

can operate in two alternative (fast and slow) gears during translation in response to the mechanical barrier of an mRNA hairpin[67]. In this slow gear, unwinding of the hairpin occurred during forward rotation of the SSU head domain suggesting that delays in steps preceeding tRNA movement into the CHI state accounted for the reduction of the overall translation speed. Ribosomes proceed in reversible sub-codon steps against the obstacle, suggesting that they exploit fluctuations on the mRNA to overcome the mechanical barrier. Our data show tRNA fluctuations between early translocation states when we replace GTP with GTPγS and in the presence of the antibiotic Spc, but these fluctuations do not increase frameshifting. Thus, it appears that there are different ways for the ribosome to switch gears, but frameshifting is promoted only at a particular step when pept-tRNA transits from CHI to P/P (Fig. 7b).

Our experiments show that at given translocation conditions, some ribosomes in a population take a fast route, whereas others switch into a slow mode. Previous molecular dynamics simulations revealed almost 500,000,000 possible kinetic sequences of intermediate states during translocation and calculated the favored route of tRNA movement[68]. Apparently, specific mRNA sequences can set the course for ribosomes to deviate from the designated route and change into alternative pathways ([67] and our study). Thus, the sequence of the mRNA appears to modulate the pace of translation not only by posing a steric hindrance for the ribosome moving along the mRNA, but also in more subtle ways that may link specific mRNA sequences with alternative translocation outcomes. The structural basis as to why the ribosomes switch to the slow gear and the physiological importance of this switch for biological processes, such as the nascent protein folding or maintenance of the mRNA stability by ribosome loading, remains to be elucidated in future work.

## Methods

All experiments were carried out in $TAKM_7$ (50 mM Tris-HCl, pH 7.5 at room temperature, 70 mM $NH_4Cl$, 30 mM KCl, and 7 mM $MgCl_2$) at 22 °C, unless stated otherwise.

Ribosomes from *E. coli*, f[³H]Met-tRNA^fMet, [¹⁴C]Gly-tRNA^Gly, [¹⁴C]Ala-tRNA^Ala, Lys-tRNA^Lys, Phe-tRNA^Phe, Val-tRNA^Val, [¹⁴C]Lys-tRNA^Lys-Cy5, initiation factors, and EF-Tu were prepared as described[50,69,70]. [¹⁴C]Gly-tRNA^Gly was prepared from *E. coli* total tRNA by selective aminoacylation of tRNA^Gly with [¹⁴C]Gly followed by isolation of [¹⁴C]Gly-tRNA^Gly in the ternary complex formed by incubating histidine-tagged EF-Tu (2-fold excess over [¹⁴C]Gly-tRNA^Gly) and GTP (1 mM), in the presence of phosphoenolpyruvate (3 mM) and pyruvate kinase (0.5%) in $TAKM_7$ for 15 min at 37 °C by affinity chromatography using Protino Ni-IDA 2000 Packed Columns (Macherey-Nagel), phenolization and ethanol precipitation of the [¹⁴C]Gly-tRNA^Gly[71]. The following mRNAs were synthesized by IBA Lifesciences and Eurofins and used for biochemical and single-molecule experiments (0-frame codons are separated by space, slippery sequence is underlined):

5'-biotin-CAACCUAAAACUUACACACCCGGCAAGGAGGUAAAUA AUG GGA AAG UUC AUUACCUAA-3'
5'-biotin-CAACCUAAAACUUACACACCCGGCAAGGAGGUAAAUA AUG UUU AAG UUC AUUACCUAA-3'
5'-biotin-CAACCUAAAACUUACACACCCGGCAAGGAGGUAAAUA AUG GCA AAA AAG UUC AUUACCUAA-3'
5'-biotin-CAACCUAAAACUUACACACCCGGCAAGGAGGUAAAUA AUG GCU AAG AAG UUC AUUACCUAA-3'

**Labeling of 70S ribosomes and tRNAs**. 70 S ribosomes labeled at protein L11 were prepared as described[23]. Protein L11 was expressed in *E. coli* BL21(DE3) after induction with isopropyl-β-D-thiogalactoside (IPTG, 1 mM). Cells were lysed by sonication in buffer (50 mM HEPES, 10 mM $MgCl_2$, 10 mM $NH_4Cl$, 1 mM dithiothreitol, 0.5 mM EDTA, pH 7.2) and inclusion bodies were solubilized in the same buffer containing 6 M urea. The protein was dialyzed against 100 volumes of the same buffer and purified by fast protein liquid chromatography (FPLC) using a HiTrap SP HP column (GE Healthcare) using a linear gradient of 10–500 mM $NH_4Cl$ in the same buffer with 6 M urea. Labeling of L11 at position C38 was carried out in the same buffer supplemented with 0.5 M $NH_4Cl$ by adding a 3-fold excess of Cy3-maleimide (GE Healthcare) dissolved in dimethylsulfoxide (DMSO) and incubating for 12 h at 4 °C. Excess dye was removed on an FLPC HiTrap SP HP column using the same salt gradient as above. To refold labeled L11, the buffer

was gradually replaced by 50 mM HEPES, 10 mM $MgCl_2$, 300 mM $NH_4Cl$, 1 mM dithiothreitol, 0.5 mM EDTA, pH 7.2, 25% glycerol using a Vivaspin 5,000 concentrator[23]. Ribosomes lacking protein L11 (70 S ΔL11) were purified from *E. coli* AM68 according to the standard protocol[69]. Reconstitution of 70 S was carried out in $TAKM_{21}$ buffer by mixing ΔL11 70 S with 10-fold excess of L11-C38-Cy3 and incubation for 45 min at 45 °C. Purification of 70S-L11-Cy3 from the excess of labeled L11 was performed by centrifugation through sucrose cushion (1.1 M sucrose in $TAKM_{21}$). Pellets were dissolved in $TAKM_7$ and the concentration was determined by absorption at 260 nm.

S13-labeled 70 S ribosomes were prepared as described[38,48]. A single-cysteine variant of protein S13 (C85S P112C) was expressed in *E. coli* BL21(DE3) after induction with IPTG (1 mM). Cells were lysed by sonication in buffer (50 mM HEPES pH 7.5, 150 mM KCl, 5% glycerol, 6 mM β-mercaptoethanol). S13 was in the insoluble fraction and inclusion bodies were dissolved in buffer (50 mM HEPES pH 7.5, 300 mM KCl, 5% glycerol, 6 mM β-mercaptoethanol, 8 M urea). Solubilized S13 was diluted in the same buffer without KCl and purified by FPLC using a HiTrap SP HP column (GE Healthcare) with a linear gradient of 50–1000 mM KCl in the same buffer with 6 M urea, followed by a Resource S column with the same KCl gradient. Cy3-labeling of S13 at C112 was carried out in the same buffer supplemented with 0.5 M $NH_4Cl$ by adding a 3-fold excess of Cy3-maleimide (GE Healthcare) dissolved in DMSO and incubating for 2 h at room temperature. Excess dye was removed on a HiTrap SP HP column using the same salt gradient as above. S13(C112-Cy3) was dialysis into 50 mM HEPES, 20 mM $MgCl_2$, 400 mM KCl, pH 7.5, 5% glycerol, 6 mM mercaptoethanol, concentrated and stored at −80 °C. 30 S subunits lacking protein S13 (30 S ΔS13) were purified from *E. coli* AM68[38]. Reconstitution of 30 S subunits was carried out in $TAKM_{20}$ buffer by mixing ΔS13 30 S with 1.5-fold excess of S13-P112C-Cy3 and incubation at 42 °C for 30 min. Purification of 30S-S13-Cy3 from the excess of labeled S13 was performed by centrifugation through a sucrose cushion (0.9 M sucrose in $TAKM_{21}$ buffer). Pellets were dissolved in $TAKM_7$ and the concentration was determined by absorption at 260 nm.

50 S subunits lacking protein L33 (50 S ΔL33) and a single-cysteine variant of protein L33 (P31C) were purified and labeled as described[38]. Reconstitution of 50 S subunits was carried out in buffer (50 mM HEPES, pH 7.5, 400 mM KCl, 20 mM $MgCl_2$, and 6 mM mercaptoethanol) by mixing ΔL33 50 S with 1.1-fold excess of L33-P31C-Cy5 and incubation for 90 min at 37 °C. 50S-L33-Cy3 were purified from the excess of labeled L33 by centrifugation through a 0.9 M sucrose cushion in $TAKM_{21}$ buffer. Pellets were dissolved in $TAKM_7$ and the concentration was determined by absorption at 260 nm.

Cy5- and BHQ2-labeling at the 3-amino-3-carboxypropyl group at uridine 47 (acp47) of tRNA^Lys was carried out by incubating purified *E. coli* tRNA^Lys with a 100-fold excess of Cy5- or BHQ2-succinimidylester (GE Healthcare and Biosearch Technologies respectively) in 50 mM HEPES, pH 8.5, for 4 h at 37 °C. Excess dye was removed by phenol extraction and ethanol precipitation. tRNA^Lys(acp47-Cy5 or -BHQ2) was aminoacylated, purified by phenol extraction and ethanol precipitation, and additionally by HPLC[23,70].

**Frameshifting assay**. To form initiation complex (IC), 70S ribosomes were incubated with a 3-fold excess of mRNA, initiation factors and f[³H]Met-tRNA^fMet and 1 mM GTP for 30 min at 37 °C. Complexes were purified by centrifugation through a 1.1 M (40%) sucrose cushion in $TAKM_{21}$ buffer (50 mM tris-HCl pH 7.5 at 37 °C, 70 mM $NH_4Cl$, 30 mM KCl, and 21 mM $MgCl_2$). Pellets were dissolved in $TAKM_7$ buffer and the concentration of purified complexes was determined by scintillation counting of f[³H]Met radioactivity. Ternary complex was prepared by incubating EF-Tu (3-fold excess over total tRNAs) with 1 mM GTP, 3 mM phosphoenolpyruvate, and 0.5% pyruvate kinase for 15 min at 37 °C and subsequent addition of aminoacyl-tRNAs (5-fold excess to 70 S) cognate to the mRNA coding sequence in 0 and −1 frame, i.e., Gly-tRNA^Gly, Lys-tRNA^Lys, Phe-tRNA^Phe and Val-tRNA^Val for the mRNA coding for fMGKF in 0 and fMGKV in −1 frame, and Ala-tRNA^Ala, Lys-tRNA^Lys, Phe-tRNA^Phe and Val-tRNA^Val for the mRNA coding for fMAKKF in 0 and fMAAKV in −1 frame.

Translation was carried out by incubating purified initiation complex (0.1 μM), ternary complexes (5-fold excess over IC) and EF-G (1 μM) with GTP or GTPγS (1 mM) at 22 °C or 37 °C for 2 min. Reaction was then quenched with KOH (0.5 M). The peptides were released by hydrolysis at 37 °C for 30 min, neutralized by adding 1/10 volume of glacial 100% acetic acid and analyzed by reversed-phase high-performance liquid chromatography (LiChrospher 100 RP-8 HPLC column, Merck) using a 0–65% acetonitrile gradient in 0.1% trifluoroacetic acid, as described[16]. The 0- and −1-frame products were quantified by scintillation counting according to f[³H]Met-tRNA^fMet and [¹⁴C]Gly-tRNA^Gly or [¹⁴C]Ala-tRNA^Ala radioactivity labels.

**Ribosome complexes for smFRET experiments**. IC and TC were prepared as described above. IC (1 μM), TC containing [¹⁴C]Gly-tRNA^Gly or [¹⁴C]Phe-tRNA^Phe (10 μM) (5 μM), and EF-G–GTP (1 μM) were mixed and incubated for 5 min at 37 °C. The resulting POST complexes were purified by centrifugation through a 1.1 M sucrose cushion in $TAKM_{21}$, the pellet was dissolved in $TAKM_7$ and the concentration was determined by [¹⁴C]Gly or [¹⁴C]Phe scintillation counting. POST complex (0.1 μM) was then incubated with a ternary complex of EF-Tu–GTP–Lys-tRNA^Lys-Cy5 (3-fold excess) in $TAKM_7$ for 5 min at 37 °C to

generate the PRE complex carrying fMGK-tRNA$^{Lys}$-Cy5 or fMFK-tRNA$^{Lys}$-Cy5 in the A site and tRNA$^{Gly}$ or tRNA$^{Phe}$ in the P site. The same procedure was followed to prepare PRE1 complexes with fMGK-tRNA$^{Lys}$-BHQ2 or fMFK-tRNA$^{Lys}$-BHQ2. The PRE complex was immobilized in the coverslip as described below.

To prepare POST complexes carrying S13-Cy3 and fMA-tRNA$^{Ala}$ in the P site, 30 S S13-Cy3 (2 μM) were incubated in TAKM$_{20}$ at 37 °C for 30 min and used to prepare initiation complexes as described above (50 S in 1.5-fold excess over 30 S, IFs, mRNA and [$^3$H]fMet-tRNA$^{fMet}$ in 3-fold excess over 30 S subunits). Ternary complexes were prepared as above using [$^{14}$C]Ala-tRNA$^{Ala}$ (10 μM). IC (1 μM), TC (5 μM) and EF-G–GTP (1 μM) were mixed together and incubated for 5 min at 37 °C. The POST complex were purified by centrifugation through a 1.1 M sucrose cushion in TAKM$_{21}$, dissolved in TAKM$_7$ and the concentration was determined by [$^{14}$C]Ala radioactivity scintillation counting. POST complex (0.1 μM) was then mixed with a 3-fold excess of EF-Tu–GTP–Lys-tRNA$^{Lys}$-Cy5, incubated for 5 min at 37 °C to generate the PRE fMAK complex, which was immobilized on the coverslip. Excess TC was removed by buffer exchange (10 volumes) with washing buffer. Subsequently, washing buffer containing EF-G–GTP (0.5 μM) was added and incubated for 5 min at RT to generate POST fMAK complex. Next, 0.3 μM unlabeled EF-Tu–GTP–Lys-tRNA$^{Lys}$ was added and incubated for 5 min at RT to form the PRE complex carrying fMAKK-tRNA$^{Lys}$ in the A site and tRNA$^{Lys}$-Cy5 in the P site.

To prepare POST complexes carrying S13-Cy3/L33-Cy5 and fMG-tRNA$^{Gly}$ or fMF-tRNA$^{Phe}$ in the P site, 30 S S13-Cy3 were incubated in TAKM$_{20}$ at 37 °C for 30 min and used to form initiation complexes as described above (50S-L33-Cy5 in 1.5-fold excess over 30 S subunits). Ternary complexes were prepared as above using [$^{14}$C]Gly-tRNA$^{Gly}$ or [$^{14}$C]Phe-tRNA$^{Phe}$. IC (1 μM), TC (5 μM), and EF-G–GTP (1 μM) were mixed together and incubated for 5 min at 37 °C. The POST complexes were purified by centrifugation through a 1.1 M sucrose cushion in TAKM$_{21}$, dissolved in TAKM$_7$ and the concentration was determined by [$^{14}$C]Gly or [$^{14}$C]Phe radioactivity scintillation counting, respectively. POST complex (0.1 μM) was then mixed with threefold excess of EF-Tu—Lys-tRNA$^{Lys}$—GTP and incubated at 37 °C for 5 min to generate the PRE complex carrying fMGK-tRNA$^{Lys}$ or fMFK-tRNA$^{Lys}$ in the A site and tRNA$^{Gly}$ or or tRNA$^{Phe}$ in the P site. The PRE complex was immobilized in the coverslip as described below.

**Coverslip preparation.** Coverslips and objective slides were sonicated in 1 M KOH for 10 min, cleaned in plasma cleaner (FEMTO plasma cleaner, Diener Electronic GmbH), silanized in 3.9 mM N1-[3-(trimethoxysilyl)propyl] diethylenetriamine (Sigma-Aldrich) and 1.7 mM acetic acid for 5 min and baked for 20 min at 110 °C. They were covered with 20 mM PEG-NHS (MeO-PEG-NHS, IRIS Biotech GmbH, PEG1165), 0.2 mM Biotin-PEG-NHS (IRIS Biotech GmbH, PEG1057) in 100 mM H$_3$BO$_3$ for 1 h at room temperature, washed with H$_2$O to remove excess of PEG, dried at 50 °C and stored under vacuum. Flow chambers were assembled by attaching cover slips on objective slides with double-sided sticky tape.

**Sample preparation and TIRF microscopy.** Purified ribosome complexes (0.1 μM) were diluted to 1 nM in buffer (TAKM$_7$ complemented with 8 mM putrescine and 1 mM spermidine). Flow chambers were incubated for 5 min at room temperature with the same buffer containing an additionally 10 mg/ml BSA and 1 mM neutravidin (Thermo Scientific) for 5 min. Neutravidin was removed by washing the flow chamber with 5-fold volume excess of the same buffer containing 1 mg/ml BSA. Ribosome complexes were added to the flow chamber and incubated for 5 min to immobilize the ribosomes on the surface through the mRNA–biotin–neutravidin interaction. Images were recorded after the addition of imaging buffer to the sample (same buffer with 2.5 mM protocatechuic acid, 50 nM protocatechuate-3,4-dioxygenase (Pseudomonas— Sigma-Aldrich), 2 mM Trolox (6-hydroxy2,5,7,8-tetramethylchromane-2-carboxylic acid) and 1 mM methylviologen (Sigma-Aldrich)). Cy3 and Cy5 fluorescence time courses during translocation were obtained by adding 0.1 μM EF-G and 1 mM GTP or GTPγS to the imaging buffer, which was added to PRE complexes immobilized on the PEG-Biotin-coated cover slips approximately 10 s before imaging. In the experiment with tRNA$^{Lys}$-BHQ2, the imaging buffer was additionally complemented with 10 nM TC containing either Phe-tRNA$^{Phe}$-Cy5 or Val-tRNA$^{Val}$-Cy5. In the experiments with fusidic acid and puromycin, 200 μM fusidic acid and 1 mM puromycin were added to the diluted sample and imaging buffer, respectively. TIRF imaging was performed on an IX 81 inverted microscope with a PLAPON 60 1.45 numerical aperture objective using xCellence rt image acquisition software (Olympus, Japan). Cy3 fluorescence was excited by a 561 nm solid-state laser, 25 mW. Images were recorded with an electron-multiplying charge-coupled device (em-CCD) camera (CCD-C9100-13, Hamamatsu, Japan). Color channels were separated by projecting donor and acceptor emission on different parts of the CCD chip using an image splitter (dual view micro imager DV2, Photometrics, USA), filter specifications HQ 605/40, HQ 680/30 (Chroma Technology). For most experiments, movies were recorded at a rate of 30.3 frames per second (33 ms per frame). Movies were recorded at a rate of 10 frame per second (100 ms per frame) in the experiment using S13-L33 FRET and EF-G(Q507D).

**Data analysis.** Fluorescence time courses for donor (Cy3) and acceptor (Cy5) were extracted using custom-made Matlab (MathWorks) software according to published protocols[23]. A semi-automated algorithm (Matlab) was used to select anti-correlated fluorescence traces exhibiting characteristic single fluorophore intensities[23]. The bleed-through of Cy3 signal into the Cy5 channel was corrected using an experimentally determined coefficient of 0.13. The FRET efficiency was defined as the ratio of the measured emission fluorescence intensities, FI$_{Cy5}$/(FI$_{Cy3}$ + FI$_{Cy5}$). Trajectories were truncated to remove photobleaching and photoblinking events. The set of all FRET traces for a given complex was compiled in a histogram, which was fitted to a sum of Gaussian functions. Matlab code using an unconstrained nonlinear minimization procedure (fminsearch, Matlab, R2011b) yields mean values and s.d. for the distribution of FRET states. Two-dimensional contour plots were generated from raw time-resolved FRET trajectories using a custom-made software. smFRET trajectories were fitted by Hidden Markov model using the vbFRET software package (http://vbfret.sourceforge.net/)[72] to generate the idealized trajectories revealing the number, sequence, and duration of FRET states. FRET changes in idealized trajectories that were smaller than the s.d. of the Gaussian distribution of the FRET states were not considered transitions because they could not be not distinguished from the noise. Dwell times of different FRET states were calculated from idealized trajectories. The dwell time distribution was fitted to an exponential function, $y = y_0 + Ae^{-t/\tau}$ to calculate the decay rate ($k = 1/\tau$). The observed rates were corrected for the photobleaching of the Cy3 and Cy5 dyes and for the observation time according to $k_{corrected} = k_{observed} - k_{photobleach} - 1/T$, where T observation time (33 s for most experiments, 100 s for S13-L33/EF-G(Q507D) experiment), $k_{photobleach} = 0.03 \pm 0.01$ s$^{-1}$. For the experiments with EF-G, FRET traces were synchronized relative to the first transition to FRET ≤ 0.1 than the lowest value of the PRE state in the absence of EF-G, if not stated otherwise. To calculate translocation rates, the time distribution between the synchronization point and the last transition to the POST state was fitted to an exponential function $y = y_0 + Ae^{-t/\tau}$. This part of the traces was also used to quantify the transition frequency, i.e., the number of transition between different states divided by the total number of transitions. GraphPad prism 8 software was used for the representation of smFRET data and fits of the data.

**Reporting summary.** Further information on research design is available in the Nature Research Reporting Summary linked to this article.

## Data availability
The data that support the findings of this study are available from the corresponding authors upon reasonable request.

## Code availability
The code used to analyze data is available from the corresponding authors upon reasonable request.

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

## Acknowledgements
We thank F. Peske for helpful discussions, B.-Z. Peng for providing the EF-G mutants, R. Belardinelli for labeled LSU, T. Senyushkina, and J.C. Thiele for help with coding. We thank O. Geintzer, V. Herold, T. Hübner, F. Hummel, S. Kappler, M. Klein, C. Kothe, A. Pfeiffer, T. Steiger and M. Zimmermann for expert technical assistance. The work was supported by the German research council (Deutsche Forschungsgemeinschaft) with project grants to M.V.R. (SFB860, project A3) and S.A. (SFB860, project A15). P.P. acknowledges an Onassis Foundation Scholarship.

## Author contributions
P.P. and A.P. performed experiments and data analysis. P.P., M.V.R., and S.A. designed experiments, interpreted the data, and wrote the paper.

## Funding

## Competing interests
The authors declare no competing interests.
