## [Peer Review File · Nature Communications]

Altered tRNA dynamics during translocation on slippery mRNA as determinant of spontaneous ribosome frameshiftingREVIEWER COMMENTS

Reviewer #1 (Remarks to the Author):

Manuscript by Poulis et al. presents additional smFRET look on the ribosome frameshifting associated with slippery sequences. Authors use EF-G mutants, different sets of slippery and non-slippery sequences, as well as slowly hydrolyzable GTP γ S, fusidic acid and spectinomycin to test previously proposed model that slow frameshifting-prone translocation mode is responsible for reduced fidelity on slippery sequences. Manuscript lacks clear advancement over previous studies, multiple control experiments (Figure 3 and 6 with wt EF-G on slippery sequence), missing data (Supplementary Fig 3 data for Q507A and Q507N are missing) as well as clarity in data presentation (ie. what is TL in figures 3 and Supplementary Fig 4?). The proposed model in Figure 7b is based on EF-G mutant Q507D data and increased population of CHI and P/P states which are almost identical in GTP γ S conditions (Fig 5), and on contrary to authors comments not really similar to spectinomycin data (in Supplementary Fig 8), which blurs authors conclusions. For these reasons I suggest that manuscript should not be published in this form.

Reviewer #2 (Remarks to the Author):

This is a very good manuscript which continues a longstanding effort by the Göttingen MPG group to fully elucidate the mechanism of -1 frameshifting. The major new results are the demonstration of two classes of ribosome during translation of slippery sequences, the correlation of the fraction of slow ribosome with frameshift efficiency, and the demonstration that tRNA dissociation from the E-site occurs more rapidly than the pep-tRNA fluctuation period. These results will be of interest to the community of researchers focused on the study of translation mechanisms. However, the points listed below need to be addressed before the MS would be acceptable for publication in Nature Communications.

MAJOR POINTS REQUIRING FURTHER CONSIDERATION:

1. Correlation of the fraction of slow ribosomes due to slippery sequences with frameshift efficiency. The results backing this conclusion are presented in Fig. 2f, for wt-EF-G and three variants. But the data referred to (Fig. S3) only include results for wt and one variant (Q507D). The results for the other two variants should also be shown in Fig. S3.

2. E-site occupancy during pep-tRNA fluctuation between CHI and P/P sites. The results presented on p. 7 clearly show that tRNA dissociation from the E-site occurs more rapidly than the fluctuation period, providing a "time window for pept-tRNA to switch to the -1-frame." This conclusion appears to be in conflict with the statement in the Discussion (p.11) that "frameshifting can occur both with one or two tRNAs bound, provided pept-tRNA is trapped in fluctuations between CHI and P/P." This apparent discrepancy should be further discussed.

3. Insufficient documentation of results presented in Figs. 3b,h. The claimed transition from CR to AC is insufficiently documented in Fig. 3b. Contour plots, number of traces are needed. Similar comments for apply to transitions from CR in Fig. 3h. The lack of detail in Fig. 3h does not sufficiently support the two sentences beginning at the bottom of p.6 and continuing into p.7

" After some time, -1-frame Val-tRNAVal-Cy5 is accommodated into POST2 and then fluctuates between A/A, A/P and A/P* states. Accommodation of Val-tRNA provides a strong evidence that after the delayed translocation the ribosome moved into the -1-frame exposing the Val codon in the A site (Fig. 3h-i)."

OTHER SIGNIFICANT POINTS:

1. A fuller description is needed of the significant structural differences between the CHI, hybrid and classical states than that currently provided in the Introduction. Given the importance of the CHI

state, this will aid the understanding of the paper by all but the most knowledgeable readers.

2. It is unclear when the EF-G is added in the TIRF microscopy experiments monitoring translocation (Figs 3b,e,h; 4c,e; 5b; 6b). This point should be clearly addressed in either the Figure legends or the Experimental section.
3. Fig. 3b – The reason for the drift in Cy3 fluorescence in PRE1(0-47 s) should be explained
4. Fig. 4 – It is not clear from the data presented that the 0.9 – 0.6 transition represents a fluctuation, as opposed to an essentially unidirectional transition from 0.9 to 0.6. Contour plots should be provided for interconversions between the 0.9, 0.6 and 0.3 FRET states
5. The current arrangement of Figures and Supplementary Figures require the reader to make frequent excursions from the main text to the Supplementary section. Could more of the Supplementary Figures be included in the main text?

MINOR EDITORIAL POINTS:

1. Last line of Discussion: Delete "the" "elucidated in the future work."
2. Fig. 3b legend: Replace "monitored" by "monitored"

Reviewer #3 (Remarks to the Author):

In this study the authors use single-molecule FRET (smFRET) to follow the translocation of tRNA^{Lys} through the ribosome in the context of a slippery mRNA sequence. By altering the placement of Cy3 and Cy5 fluorophores, they are able to follow the dynamics of EF-G catalyzed translocation relative to translocation on a non-slippery mRNA as a control. They show that translocation on a slippery sequence can proceed through two pathways. In one pathway, both A and P site tRNAs rapidly move to the P and E sites with no change in reading frame. In the other pathway the deacylated P-site tRNA rapidly translocates while the A-site peptidyl-tRNA is delayed at a late chimeric stage of translocation, where it fluctuates between chimeric (ap/P) and post-translocation (P/P) states. During this period the authors confirm that the small subunit head is swiveled, conditions where codon-anticodon pairing is destabilized and alternative pairing frames can be explored. In the case of spontaneous –1 frameshifting, rapid release of deacylated tRNA from the E-site following translocation may facilitate –1 pairing of the stalled peptidyl-tRNA. These insights into translocation on a slippery sequence help to explain how spontaneous –1 frameshifting occurs.

Central to this study is the demonstration of a positive correlation of –1 frameshifting on a slippery mRNA sequence to the fraction of peptidyl-tRNA that is delayed in translocation. This was accomplished by carrying out translocation in the presence of several mutants of EF-G containing amino acid substitutions at position 507. In the wild-type factor, glutamine at this position helps to stabilize codon-anticodon pairing of the peptidyl-tRNA. Substitutions of this amino acid are known to promote –1 frameshifting and the authors show that these same substitutions increase the fraction of translocation that is delayed due to fluctuation between ap/P and P/P states.

The smFRET studies appear to be carefully done. However, several minor revisions are needed.

1. As regards to the biochemical assay for –1 frameshifting, it is not specified whether peptide formation was carried out in the presence of both tRNA^{Phe} and tRNA^{Val} or only in the presence of the latter. By conducting this type of assay in the presence of both the 0-frame and -1-frame incoming tRNAs, any possibility of frameshifting in the P-site can be discounted.
2. The incorporation of 0- and –1 frame aa-tRNAs presented in Figure 3 does not add a lot to the paper and could be included in the supplemental section.
3. It is not clear from the methods section (pg 15) how the POST fMAK complex is formed without going on to give a POST fMAKK complex. This needs to be more clearly described.
4. The findings of the work are consistent with recent work that elucidated the mechanism of spontaneous +1 frameshifting. In particular, the findings that –1 frameshifting occurs at a later stage

of the translocation reaction, involving the head domain swiveling of the small subunit, are reminiscent of the findings for spontaneous +1 frameshifting as reported in PMID: 33436566 and in PMID: 34330903. Both of these papers should be cited and discussed in the present manuscript. This will give readers a broader and more comprehensive view of the background of this work.

Answer to the REVIEWER COMMENTS

Reviewer #1 (Remarks to the Author):

Manuscript by Poulis et al. presents additional smFRET look on the ribosome frameshifting associated with slippery sequences. Authors use EF-G mutants, different sets of slippery and non-slippery sequences, as well as slowly hydrolyzable GTP γ S, fusidic acid and spectinomycin to test previously proposed model that slow frameshifting-prone translocation mode is responsible for reduced fidelity on slippery sequences.

Manuscript lacks clear advancement over previous studies, multiple control experiments (Figure 3 and 6 with wt EF-G on slippery sequence), missing data (Supplementary Fig 3 data for Q507A and Q507N are missing) as well as clarity in data presentation (ie. what is TL in figures3 and Supplementary Fig 4?).

Reply: We politely disagree with the referee concerning the novelty and originality of our study. While previous work (reviewed in detail in the Introduction) has identified the putative link between the translocation rate and the tendency for the ribosome to frameshift, in this work we reveal the molecular mechanism on the single molecule level and additionally dissect the timing and the exact coupling between ribosome motions and frameshifting. Notably, in the previous work on programmed ribosome frameshifting slow translocation was induced by the downstream secondary structure element on the mRNA (which is not very surprising, as the ribosome encounters a hurdle that it struggles to pass). In contrast, here we study spontaneous frameshifting without such hurdles and show that an mRNA slippery sequence by itself changes the choreography of translocation, leading to frameshifting.

One of the control experiments “Figure 6 with wt EF-G on slippery sequence” requested by the referee was already present in our original text (Fig. 6a,b). We now performed the second control experiment concerning “Figure 3 with wt EF-G on slippery sequence”; the data are presented in the revised Fig. 3 and Supplementary Fig. 6 and 7. We show that also with EF-G(wt) ribosomes show slow translocation of pept-tRNA on slippery mRNA before accommodation of -1 -frame Val-tRNA^{Val} (Fig. 3e-h). However, since the majority of ribosomes stays in 0-frame with EF-G(wt) (Fig. 3a-d), -1 -frame Val-tRNA^{Val} is predominantly rejected (Fig. 3a-d). We never observed accommodation of -1 -frame Val-tRNA^{Val} after fast translocation of pept-tRNA, which makes prolonged fluctuations between CHI and P/P states an essential determinant of spontaneous frameshifting. We worked extensively on the data presentation in Fig. 3 and Supplementary Fig. 4, 5, 6, and 7. We simplified the cartoons, explained all abbreviations in the figure legends and replaced the term “TL” (translocation) for “CHI state”, which was explained in the Introduction. The revised versions of Fig. 3 and Supplementary Fig.

4, 5,6, and 7 show that slow translocation via CHI and P/P states is a major determinant of spontaneous ribosome frameshifting.

We also added the data on the frameshifting efficiency by EF-G(Q507A) and EF-G(Q507N) mutants in Supplementary Fig. 3.

The proposed model in Figure 7b is based on EF-G mutant Q507D data and increased population of CHI and P/P states which are almost identical in GTP γ S conditions (Fig 5), and on contrary to authors comments not really similar to spectinomycin data (in Supplementary Fig 8), which blurs authors conclusions. For these reasons I suggest that manuscript should not be published in this form.

Reply: We are surprised by this comment of the referee, because our transition frequency analysis clearly shows differences in translocation in the presence of EF-G(Q507D) and EF-G—GTP γ S (compare Fig. 2e and Fig. 5d). EF-G(Q507D) fluctuates predominantly between FRET 0.4 and 0.2, whereas EF-G—GTP γ S fluctuates between FRET 0.6 and 0.4, i.e. EF-G is stalled at a different step of translocation. The frameshifting efficiency is high during translocation with EF-G(Q507D) and pept-tRNA fluctuates between CHI and P/P states (Fig. 2e and Supplementary Fig. 3). In contrast, the frameshifting efficiency during translocation by EF-G—GTP γ S is low and tRNAs fluctuate predominantly between CHI and A/P* states (Fig. 5d). These results show that frameshifting occurs while ribosomes sample CHI and P/P states, whereas sampling of earlier intermediate states does not lead to frameshifting. Presumably, the referee is confused by the fact that contour plots with EF-G(Q507D) and EF-G—GTP γ S look similar on first sight (Fig. 2d and Fig. 5c). This is because these plots are dominated by the end levels (the P/P state) and were therefore further analyzed with respect to the transition frequency (Fig. 2e). In this analysis, we counted the number of transitions between FRET 0.6 (A/A, A/P*), FRET 0.4 (CHI) and 0.2 (P/P) states after EF-G binding to PRE complexes using idealized traces derived from the HMM fit of the FRET time course (see Methods). This information reveals in which step of the translocation pathway the movement of pept-tRNA is stalled (see also Adio et al. 2015). We added information in the Method section to better explain how the transition frequencies were derived.

The translocation pathway by EF-G—GTP γ S is similar to that observed with EF-G(wt)—GTP in the presence of Spc (Fig. 5 and Supplementary Fig.11), as translocation is stalled at an early stage, consistent with other existing literature (Rundlet et al., Nature 2021, Belardinelli et al., RNA 2019 and Adio et al., Nat Commun 2015). In both cases, pept-tRNA is stalled in fluctuations between A/P, A/P* and CHI states and transitions between CHI and P/P are practically absent. Translocation of deacylated tRNA is slow and occurs at the same rate as the translocation of pept-tRNA. We observed exactly the opposite behavior by EF-G(Q507D) (Fig. 2e, Fig.4e,f), where pept-tRNA is trapped in fluctuations between CHI and P/P, and translocation of deacylated tRNA is fast.

Our model (Fig. 7) shows the uncoupled movement of pept- and deacylated tRNA observed with the frameshifting-prone EF-G(Q507D) mutant. EF-G—GTPyS and Spc show only background levels of frameshifting (Fig. 5a), which is why these conditions serve as relevant controls but should not enter the main model.

Reviewer #2 (Remarks to the Author):

This is a very good manuscript which continues a longstanding effort by the Göttingen MPG group to fully elucidate the mechanism of -1 frameshifting. The major new results are the demonstration of two classes of ribosome during translation of slippery sequences, the correlation of the fraction of slow ribosome with frameshift efficiency, and the demonstration that tRNA dissociation from the E-site occurs more rapidly than the pep-tRNA fluctuation period. These results will be of interest to the community of researchers focused on the study of translation mechanisms.

However, the points listed below need to be addressed before the MS would be acceptable for publication in Nature Communications.

MAJOR POINTS REQUIRING FURTHER CONSIDERATION:

1. Correlation of the fraction of slow ribosomes due to slippery sequences with frameshift efficiency. The results backing this conclusion are presented in Fig. 2f, for wt-EF-G and three variants. But the data referred to (Fig. S3) only include results for wt and one variant (Q507D). The results for the other two variants should also be shown in Fig. S3.

Reply: We added the data on the frameshifting efficiency by EF-G(Q507A) and EF-G(Q507N) mutants in Supplementary Fig. 3.

2. E-site occupancy during pep-tRNA fluctuation between CHI and P/P sites. The results presented on p. 7 clearly show that tRNA dissociation from the E-site occurs more rapidly than the fluctuation period, providing a “time window for pept-tRNA to switch to the -1-frame.” This conclusion appears to be in conflict with the statement in the Discussion (p.11) that “frameshifting can occur both with one or two tRNAs bound, provided pept-tRNA is trapped in fluctuations between CHI and P/P.” This apparent discrepancy should be further discussed.

Reply: We show that upon spontaneous -1-frameshifting, deacylated tRNA dissociates rapidly from the E site while pept-tRNA is still in the process of (slow) translocation (Fig. 1, 3 and 4). This indicates a one-tRNA slippage mechanism because the E-site codon is unoccupied while pept-tRNA has not reached the POST state. In contrast, during -1PRF (described in the discussion on p.12) the translocation of deacylated and pept-tRNA is coupled and occurs at similar rate (Caliskan et al., Cell 2014). Hence, depending on whether frameshifting occurs

spontaneously or in a programmed manner, it can follow a one- or two-tRNA slippage mechanism. We modified the text (see top of p. 12) to point out that the two-tRNA slippage corresponds to -1 PRF.

3. Insufficient documentation of results presented in Figs. 3b,h. The claimed transition from CR to AC is insufficiently documented in Fig. 3b. Contour plots, number of traces are needed. Similar comments for apply to transitions from CR in Fig. 3h. The lack of detail in Fig. 3h does not sufficiently support the two sentences beginning at the bottom of p.6 and continuing into p.7

“ After some time, -1 -frame Val-tRNA^{Val}-Cy5 is accommodated into POST2 and then fluctuates between A/A, A/P and A/P* states. Accommodation of Val-tRNA provides a strong evidence that after the delayed translocation the ribosome moved into the -1 -frame exposing the Val codon in the A site (Fig. 3h-i).”

Reply: We revised Fig. 3 and the corresponding figure legend. We explain the meaning of the abbreviation “CR” (codon recognition) and show that upon accommodation in the A site, tRNA^{Phe} and tRNA^{Val} fluctuate between classic (A/A) and hybrid (A/P and A/P*) conformations in a similar manner as pept-tRNA^{Lys} (Supplementary Fig. 1). We included n=number of traces in the figure legend and compiled FRET signals reporting on the interaction of aa-tRNAs with POST2 complexes into contour plots (Fig. 3d,h,l and Supplementary Fig. 5d,h). The dwell time analysis of these FRET signals is now presented in the new Supplementary Fig. 6. The dissociation rates (k_{off}) differ dramatically for the cognate aa-tRNA (which is accommodated) and the near-cognate tRNA (which is rapidly is rejected). Dissociation of -1 -frame tRNA^{Val} from PRE2 complexes formed after slow translocation on slippery mRNA is as slow as of the cognate 0-frame tRNA^{Phe} on PRE2 complexes formed on non-slippery mRNA. This provides a strong evidence in favor of our conclusion that slowly-translocating ribosomes have moved into the -1 -frame exposing the Val codon in the A site.

Please note that we also unified the experiment schemes in Fig. 3a, e, i. In all experiments, PRE1 complexes are immobilized and EF-G (wt or mutant) is added together with EF-Tu-aa-tRNA-GTP complex to induce tRNA accommodation and subsequent translocation.

OTHER SIGNIFICANT POINTS:

1. A fuller description is needed of the significant structural differences between the CHI, hybrid and classical states than that currently provided in the Introduction. Given the importance of the CHI state, this will aid the understanding of the paper by all but the most knowledgeable readers.

Reply: We have added the description of the dynamics of PRE translocation complexes including explanation of tRNA conformations in hybrid and classical states. We have further

explained that the CHI state is a transient intermediate state of translocation, which rapidly resolves into the POST state.

2. It is unclear when the EF-G is added in the TIRF microscopy experiments monitoring translocation (Figs 3b,e,h; 4c,e; 5b; 6b). This point should be clearly addressed in either the Figure legends or the Experimental section.

Reply: In the experiments described in Fig. 3, EF-G was added to immobilized PRE1 complexes together with EF-Tu-GTP-aa-tRNA complex. The components were added with the imaging buffer approximately 10 s prior to imaging. We added additional description in the figure legends and also in the Methods section.

In the experiments described in Fig. 4, 5, 6 EF-G was added to immobilized PRE complexes approximately 10 s before imaging. This is now clearly stated in the Methods section.

Please note that the vertical line in Fig. 6 represents the synchronization point of the smFRET traces. This is now explained in the figure legend.

3. Fig. 3b – The reason for the drift in Cy3 fluorescence in PRE1(0-47 s) should be explained.

Reply: “The drift” in the Cy3 fluorescence signal of the example trace shown in our previous Fig. 3b represented noise in the data and did not correspond to tRNA movement. We replaced the trace for a different example trace with stable Cy3 fluorescence.

4. Fig. 4 – It is not clear from the data presented that the 0.9 – 0.6 transition represents a fluctuation, as opposed to an essentially unidirectional transition from 0.9 to 0.6. Contour plots should be provided for interconversions between the 0.9, 0.6 and 0.3 FRET states

Reply: We changed the text (p. 7) to clarify that the transitions between 0.9 and 0.6 states are not unidirectional but much less frequent than fluctuations of pept-tRNA on PRE complexes. Additionally, we calculated the transition frequency of deacylated tRNA (i.e. the average number of transitions between 0.6 and 0.9 FRET per trace) and compared it with the transition frequency of pept-tRNA on PRE complexes. Values differ by about one order of magnitude (0.4 vs 5.4 transitions, Supplementary Fig. 1c and 8f), further supporting the point that fluctuations of deacylated tRNA on PRE complexes are relatively slow. Transitions into the POST (0.3 FRET) state are irreversible (Supplementary Fig. 9d) and occur from the PRE state with either FRET 0.9 or 0.6, which is shown in the contour plot (Fig. 4d,e).

5. The current arrangement of Figures and Supplementary Figures require the reader to make frequent excursions from the main text to the Supplementary section. Could more of the

Supplementary Figures be included in the main text?

Reply: In the current version of the manuscript, we describe all key experiments and the model in the main text and present the control experiments in the supplement. We absolutely see the point made by the referee, but the main Figs are already heavy on panels and information. We fear that including even more material into the main text will lead to overload and confusion. We are therefore reluctant to change the principle sequence of figures in the main text.

MINOR EDITORIAL POINTS:

1. Last line of Discussion: Delete “the” “elucidated in the future work.”

Corrected in the revised manuscript.

2. Fig. 3b legend: Replace “monitored’ by “monitored”

This is corrected in the revised manuscript.

Reviewer #3 (Remarks to the Author):

In this study the authors use single-molecule FRET (smFRET) to follow the translocation of tRNA^{Lys} through the ribosome in the context of a slippery mRNA sequence. By altering the placement of Cy3 and Cy5 fluorophores, they are able to follow the dynamics of EF-G catalyzed translocation relative to translocation on a non-slippery mRNA as a control. They show that translocation on a slippery sequence can proceed through two pathways. In one pathway, both A and P site tRNAs rapidly move to the P and E sites with no change in reading frame. In the other pathway the deacylated P-site tRNA rapidly translocates while the A-site peptidyl-tRNA is delayed at a late chimeric stage of translocation, where it fluctuates between chimeric (ap/P) and post-translocation (P/P) states. During this period the authors confirm that the small subunit head is swiveled, conditions where codon-anticodon pairing is destabilized and alternative pairing frames can be explored. In the case of spontaneous -1 frameshifting, rapid release of deacylated tRNA from the E-site following translocation may facilitate -1 pairing of the stalled peptidyl-tRNA. These insights into translocation on a slippery sequence help to explain how spontaneous -1 frameshifting occurs.

Central to this study is the demonstration of a positive correlation of -1 frameshifting on a slippery mRNA sequence to the fraction of peptidyl-tRNA that is delayed in translocation. This was accomplished by carrying out translocation in the presence of several mutants of EF-G containing amino acid substitutions at position 507. In the wild-type factor, glutamine at this position helps to stabilize codon-anticodon pairing of the peptidyl-tRNA. Substitutions of this

amino acid are known to promote -1 frameshifting and the authors show that these same substitutions increase the fraction of translocation that is delayed due to fluctuation between ap/P and P/P states.

The smFRET studies appear to be carefully done. However, several minor revisions are needed.

1. As regards to the biochemical assay for -1 frameshifting, it is not specified whether peptide formation was carried out in the presence of both tRNA^{Phe} and tRNA^{Val} or only in the presence of the latter. By conducting this type of assay in the presence of both the 0-frame and -1 -frame incoming tRNAs, any possibility of frameshifting in the P-site can be discounted.

Reply: The biochemical assays to quantify frameshifting efficiency (Fig. 2f, 5a, and Supplementary Fig.3) were conducted in the presence of equal concentrations of the 0-frame Phe-tRNA^{Phe} and -1 -frame Val-tRNA^{Val} (5-fold excess over 70S IC). We clarified the description of the experimental procedure in the Methods section (p. 16).

2. The incorporation of 0- and -1 frame aa-tRNAs presented in Figure 3 does not add a lot to the paper and could be included in the supplemental section.

Reply: We feel that we cannot omit Fig. 3 from the main text figures because Fig 2 and Fig 3 represent two different aspects of pept-tRNA^{Lys} translocation. While Fig 2 reveals that pept-tRNA samples distinct intermediate states of translocation (CHI and POST) during spontaneous frameshifting, experiments presented in Fig. 3 verify that slow translocation via CHI states indeed precedes the accommodation of -1 -frame-tRNA. Without this direct evidence, the correlation between slow translocation via distinct CHI states and spontaneous frameshifting would remain a conjecture.

3. It is not clear from the methods section (pg 15) how the POST fMAK complex is formed without going on to give a POST fMAKK complex. This needs to be more clearly described.

Reply: We revised the corresponding section in Methods (p. 17). Now it contains all the essential steps for the sample preparation.

4. The findings of the work are consistent with recent work that elucidated the mechanism of spontaneous $+1$ frameshifting. In particular, the findings that -1 frameshifting occurs at a later stage of the translocation reaction, involving the head domain swiveling of the small subunit, are reminiscent of the findings for spontaneous $+1$ frameshifting as reported in PMID: 33436566 and in PMID: 34330903. Both of these papers should be cited and discussed in the

present manuscript. This will give readers a broader and more comprehensive view of the background of this work.

Reply: We agree that comparison of our results on spontaneous -1 -frameshifting with spontaneous $+1$ -frameshifting gives a broader view on the background of our study. We added a new paragraph on this subject into the Discussion section highlighting mechanistic similarities and differences between the two pathways.

REVIEWERS' COMMENTS

Reviewer #1 (Remarks to the Author):

The manuscript by Poulis et al., has improved in the representation of data and additional controls are included. The discussion of the data is also clarified and additional explanations are provided.

Reviewer #2 (Remarks to the Author):

The authors have responded well to my concerns. A few points still remain, as listed below.

Significant points:

1. Although a CR state is a plausible intermediate on the way to the PRE2 complex, the results interpreted as demonstrating its formation (Figs.3 h,i; Supplementary Fig. 5d) are not compelling, in part due to the small number of traces obtained. As a result, the current text (bottom, p. 6) is overstated, and should be reworded more conservatively.

2. In a similar vein, on p.7, line 3 from the bottom, the word "show" should be replaced by "suggest".

Minor points:

1. Supplementary Figure 3 legend needs to be edited to reflect the changes in part (a)

2. Three edits to p. 12, para2

- line 2: post-transcriptional

- line 8: replace "any type of frameshifting" with "-1 or +1 frameshifting"

-line 10: stalls

Reviewer #3 (Remarks to the Author):

The revisions by the authors have addressed all of my concerns. more comprehensive view of the background of this work.

Point-by-point reply

Reviewer #1 (Remarks to the Author):

The manuscript by Poulis et al., has improved in the representation of data and additional controls are included. The discussion of the data is also clarified and additional explanations are provided.

Reply: We appreciate the positive feedback by Reviewer #1 and thank him/her for the helpful remarks on the initial submission.

Reviewer #2 (Remarks to the Author):

The authors have responded well to my concerns. A few points still remain, as listed below.

Significant points:

1. Although a CR state is a plausible intermediate on the way to the PRE2 complex, the results interpreted as demonstrating its formation (Figs.3 h,l; Supplementary Fig. 5d) are not compelling, in part due to the small number of traces obtained. As a result, the current text (bottom, p. 6) is overstated, and should be reworded more conservatively.

Reply: We replaced “CR” for codon recognition for “RB” for ribosome binding state, where Phe-tRNA^{Phe}-Cy5 reads the codon in the A site according to the step assignment of the earlier work (Geggier et al., 2010).

2. In a similar vein, on p.7, line 3 from the bottom, the word “show” should be replaced by “suggest”.

Reply: Done

Minor points:

1. Supplementary Figure 3 legend needs to be edited to reflect the changes in part (a)

Reply: We changed the legend of Supplementary Fig. 3.

2. Three edits to p. 12, para2

- line 2: post-transcriptional

- line 8: replace “any type of frameshifting” with “-1 or +1 frameshifting”

-line 10: stalls

Reply: We corrected the typos in the text.

Reviewer #3 (Remarks to the Author):

The revisions by the authors have addressed all of my concerns.

Reply: We appreciate the positive response to the revised manuscript.